# Searching for ultralight dark matter conversion in solar corona using Low Frequency Array data

Haipeng An [1,2,3,4] ✉, Xingyao Chen[5] ✉, Shuailiang Ge [3,6] ✉, Jia Liu [3,6] ✉ & Yan Luo [6] ✉

Ultralight dark photons and axions are well-motivated hypothetical dark matter candidates. Both dark photon dark matter and axion dark matter can resonantly convert into electromagnetic waves in the solar corona when their mass is equal to the solar plasma frequency. The resultant electromagnetic waves appear as monochromatic signals within the radio-frequency range with an energy equal to the dark matter mass, which can be detected via radio telescopes for solar observations. Here we show our search for converted monochromatic signals in the observational data collected by the high-sensitivity Low Frequency Array (LOFAR) telescope and establish an upper limit on the kinetic mixing coupling between dark photon dark matter and photon, which can reach values as low as $10^{-13}$ within the frequency range of 30 – 80 MHz. This limit represents an improvement of approximately one order of magnitude better than the existing constraint from the cosmic microwave background observation. Additionally, we derive an upper limit on the axion-photon coupling within the same frequency range, which is better than the constraints from Light-Shining-through-a-Wall experiments while not exceeding the CERN Axion Solar Telescope (CAST) experiment or other astrophysical bounds.

Due to the absence of significant results in the search for weakly interacting massive particles (WIMPs)[1–3], increasing attention has shifted towards the ultralight dark matter (DM) candidates, including dark photons, quantum chromodynamic (QCD) axions and axion-like particles. Dark photon is a hypothetical vector ultralight DM candidate[4–7], constituting one of the simplest extensions of the Standard Model (SM) by incorporating a massive vector field coupled to the photon field through the kinetic mixing marginal operator[8–13]. There are several ways to produce the right amount of dark photon dark matter (DPDM) during the early Universe, including the misalignment mechanism with a non-minimal coupling to the Ricci scalar[5,6,14–16], inflationary fluctuations[7,17–26], parametric resonances[27–32],

or the decay of the cosmic strings[33]. The QCD axion, initially introduced to address the strong CP problem as a hypothetical particle[34–37], where "CP" stands for the combination of charge conjugation symmetry and parity symmetry, has been shown to be a good DM candidate[38]. Axion-like particles arising in, e.g., string-theory models[39], coupled to SM particles in a similar way, also stand as promising DM candidates. Axions or axion-like particles can be generated by the misalignment mechanism[40–42], or the decay of topological objects[43,44] during the early Universe.

The couplings between dark photon or axions and SM particles provide important tools in searching for these ultralight particles. Various types of experiments are looking for the signals associated

[1]Department of Physics, Tsinghua University, 100084 Beijing, China. [2]Center for High Energy Physics, Tsinghua University, 100084 Beijing, China. [3]Center for High Energy Physics, Peking University, 100871 Beijing, China. [4]Frontier Science Center for Quantum Information, 100084 Beijing, China. [5]School of Physics & Astronomy, University of Glasgow, Glasgow G12 8QQ, UK. [6]School of Physics and State Key Laboratory of Nuclear Physics and Technology, Peking University, 100871 Beijing, China. ✉e-mail: anhp@mail.tsinghua.edu.cn; Xingyao.Chen@glasgow.ac.uk; sge@pku.edu.cn; jialiu@pku.edu.cn; ly23@stu.pku.edu.cn

with photons, including haloscopes for Galactic halo DM[45,46], helioscopes for ultralight particles emitted from the Sun[45,46], and the "Light Shining through the wall" (LSW) methods[47,48]. Dark photons and axions can also be detected via WIMP detectors[49,50]. Moreover, many experimental results initially intended for axion DM can be reinterpreted for dark photons. A comprehensive summary of experimental constraints (including projected ones) for dark photons and axions can be found in ref. 51,52.

The other meaningful way to look for axions or dark photons is to investigate anomalous signals in various astrophysical environments, such as the cosmic microwave background (CMB) spectral distortion constraints on dark photons[6,53], gamma-ray constraints on axion DM[54,55], neutron stars[56–67], white dwarfs[66,68–70], supernovae[71,72], quasars and blazars[73–78], the Sun, red giants and horizontal branch stars[79–81], and globular clusters[82,83]. These searches assume that the ultralight particles are either DM or sourced inside the astrophysical objects. Remarkably, the Sun, being our closest star, offers a good laboratory for probing ultralight particles. Previous works have set constraints on ultralight particles generated inside the Sun via stellar cooling[79,80,84] and axion decay[85]. On the other hand, ref. 86 proposed that DPDM can resonantly convert into monochromatic radio-frequency electromagnetic (EM) waves in the solar corona. This phenomenon occurs at a radius where the plasma frequency equals the DPDM mass. Furthermore, with the presence of the solar magnetic field, axion DM can also resonantly convert into radio waves in the solar corona.

In this work, we investigate such resonantly converted monochromatic radio signal within the solar observation data collected by Low Frequency Array (LOFAR) telescope[87]. To calculate the signal, we carry out simulations of EM wave propagation inside solar corona of the quiet Sun. Subsequently, we compare the signal with the LOFAR data to deduce the upper limits for both the DPDM model and axion DM model. However, due to the relatively weak nature of the solar coronal magnetic field, our constraint on axion parameters does not exceed many existing constraints. Therefore, we focus on the dark photon case in the main text while leaving the detailed discussion of the axion case in the methods, subsection Constraint on axion-like particle dark matter. We set the 95% C.L. upper limit on the kinetic mixing coupling between DPDM and photon to about $10^{-13}$ within the frequency range of 30−80 MHz.

## Results

### Resonant conversion of ultralight DM into photons in solar plasma

For the DPDM model, dark photons interact with SM particles through kinetic mixing, and the corresponding Lagrangian can be written as

$$\mathcal{L}_{A'\gamma} = -\frac{1}{4}F'_{\mu\nu}F'^{\mu\nu} + \frac{1}{2}m_{A'}^2 A'_\mu A'^\mu - \frac{1}{2}\epsilon F_{\mu\nu}F'^{\mu\nu}, \quad (1)$$

where $A'$ and $\gamma$ represent dark photon and photon respectively, $F_{\mu\nu}$ and $F'^{\mu\nu}$ represent the field strengths of photon and dark photon respectively, with the Greek letters $\mu, \nu$ denoting the vector indices, $m_{A'}$ denotes the mass of dark photon, $A'_\mu$ is the vector field of dark photon, and $\epsilon$ stands for the kinetic mixing parameter.

In the solar corona, the presence of free electrons gives rise to a plasma frequency denoted by $\omega_p$, serving as the effective mass for the EM wave. This quantity is determined by the free electron density $n_e$ in the non-relativistic plasma, and can be represented as

$$\omega_p = \left(\frac{4\pi\alpha_{EM}n_e}{m_e}\right)^{\frac{1}{2}} = \left(\frac{n_e}{7.3\times10^8 \text{cm}^{-3}}\right)^{\frac{1}{2}}\mu\text{eV}, \quad (2)$$

where $\alpha_{EM}$ represents the fine structure constant and $m_e$ is the electron mass. It is noteworthy that we employ natural units throughout our paper, thereby setting $\hbar$ and $c$ to unity: $\hbar = c = 1$. When a dark photon $A'$

propagates in the plasma, it can resonantly convert into a SM photon when $\omega_p \approx m_{A'}$[86]. In the solar corona, we have $n_e$ monotonically decreasing from $10^{10}$ to $10^6$ cm$^{-3}$ with increasing height above the solar photosphere. Therefore, the corresponding plasma frequency scans from $4\times10^{-6}$ to $4\times10^{-8}$ eV. If the DM mass $m_{A'}$ falls within this range, the resonant conversion of DPDM into EM waves can occur at a specific radius $r_c$ satisfying $\omega_p(r_c) = m_{A'}$. The frequency of the converted EM wave, $m_{A'}/(2\pi)$, lies within the radio-frequency range of about 10−1000 MHz. Therefore, it can be tested by various radio telescopes engaged in solar physics programs, such as LOFAR[87] and SKA[88]. Since DM in the Galactic halo is non-relativistic with the typical velocity $v_{DM}$ approximately $10^{-3}$ times the speed of light, the converted EM wave is nearly monochromatic with a spread of about $10^{-6}$ around its central value[86]. The DM dispersion bandwidth $B_{sig}$ can be evaluated by

$$B_{sig} \approx \frac{m_{A'}v_{DM}^2}{2\pi} \approx 130\,\text{Hz}\left(\frac{m_{A'}}{\mu\text{eV}}\right). \quad (3)$$

Our analysis adopts the electron density profile for the quiet Sun provided by LOFAR observations[89], shown as the solid blue line in Fig. 1, and the DM wind constantly passes through the solar atmosphere. For specific details regarding different solar density profiles in the context of the quiet Sun, refer to the methods, subsection The solar model. The probability of DPDM resonantly converting into photons is[86,90]

$$P_{A'\to\gamma}(v_{rc}) = \frac{2}{3}\times\pi\epsilon^2 m_{A'}v_{rc}^{-1}\left|\frac{\partial\ln\omega_p^2(r)}{\partial r}\right|_{r=r_c}^{-1}, \quad (4)$$

where $v_{rc}$ is the radial velocity at the resonant layer. The prefactor 2/3 arises in Eq. (4) because the longitudinal mode of photons converted from the corresponding mode of dark photon cannot propagate out of the plasma. The conversion probability Eq. (4) accounts for two transverse modes, and we assume that dark photons polarize in three directions with equal probability.

Utilizing the conversion probability, the radiation power $\mathcal{P}$ per solid angle $d\Omega$ at the conversion layer can be derived as

$$\frac{d\mathcal{P}}{d\Omega} = \int d\mathbf{v}_0 f_{DM}(\mathbf{v}_0) P_{A'\to\gamma}(v_0)\rho_{DM}v(r_c)r_c^2, \quad (5)$$

where the DM density is $\rho_{DM} = 0.3$ GeV cm$^{-3}$[91,92] and $\mathbf{v}_0$, the initial DM velocity, follows a Maxwellian distribution $f_{DM}(\mathbf{v}_0)$ with the most

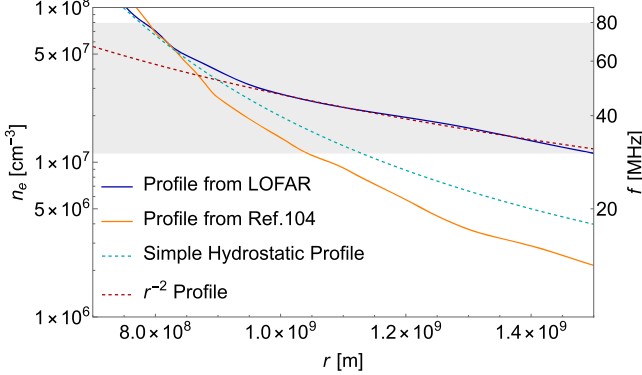

**Fig. 1 | Comparison between different electron density profiles.** Various density profiles are depicted using different lines. The solid blue line represents the profile derived from LOFAR observations[89]. In comparison, the solid orange line represents the profile from ref. 104. The dashed cyan line represents a simple hydrostatic model and the dashed red line represents an $r^{-2}$ profile[89]. The gray shaded region denotes the frequency region -30−80 MHz which our study focuses on.

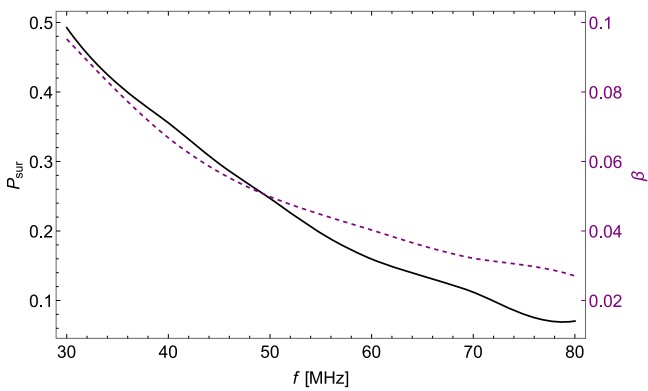

**Fig. 2 | The propagation coefficients as functions of frequency.** The survival probability $P_{sur}$ and the smearing factor $\beta$ as functions of the photon frequency $f$ are depicted as the solid black line and the solid purple line, respectively.

probable velocity of 235 km/s[93,94]. $v(r_c) = \sqrt{v_0^2 + 2G_N M_\odot / r_c}$ corresponds to the DM velocity at the conversion layer including the gravitational effect of the Sun, with $G_N$ standing for the gravitational constant and $M_\odot$ representing the solar mass. Detailed derivations for the conversion probability and the radiation power can be found in the methods, subsection The conversion probability of $A' \to \gamma$ and the radiation power. We have furthermore demonstrated that electron density fluctuations do not change the result of the conversion probability in the methods, subsection Impact of small-scale fluctuations on conversion probability.

**Propagation of converted photons in solar plasma**

The converted EM waves propagating through the corona will experience interactions with the plasma, including both absorption and scattering processes. The absorption of these converted photons is mainly through the inverse bremsstrahlung process. Due to refraction, the converted EM wave would propagate radially outward once it exits of the resonant region if scatterings between the EM wave and the plasma were absent[86]. However, the presence of scattering in the inhomogeneous plasma will randomize the direction of EM waves, leading to a broadened angular distribution of the outgoing EM waves[95,96]. We are using LOFAR data made in the tied-array beam mode. While this mode offers a nice angular resolution[87], the field of view (FOV) of each LOFAR beam is significantly smaller than the total angular span of the Sun. Consequently, we expect the scattering effect to suppress the signal observed by the LOFAR detector.

When accounting for both absorption and scattering effects, the spectral flux density received by LOFAR can be expressed as

$$S_{sig} = \frac{1}{\mathcal{B}} \frac{1}{d^2} \frac{d\mathcal{P}}{d\Omega} P_{sur}(f) \beta(f), \tag{6}$$

where $d = 1$ AU is the distance between Earth and the Sun. $\mathcal{B}$ represents the bandwidth, which is the larger one between the DM dispersion bandwidth $B_{sig}$ which is about 130 Hz and the spectral resolution of the telescope $B_{res} = 97$ kHz. In our case, $B_{sig} \ll B_{res}$, so we have $\mathcal{B} = B_{res}$. The survival probability $P_{sur}(f)$ and the factor $\beta(f)$ are defined later. It is noteworthy that the energy dispersion could be enlarged by scatterings with the plasma inhomogeneities. However, this impact is negligible because the inhomogeneities can be treated as effectively static, given their velocities are much lower than the speed of light, and only elastic scatterings need to be considered[96]. The speed of inhomogeneities may become important for photons with the smallest velocities just after conversion. The typical density fluctuation is the ion-sound waves[97] with the speed $C_s \approx \sqrt{[T_e(1 + 3T_i/T_e)/m_i]}$ which is about 100 km/s ($T_e$, $T_i$, and $m_i$ are respectively the electron temperature, ion temperature, and ion mass), which is comparable with the DM velocity

$v_{DM}$ which is approximately $10^{-3}c$ in Eq. (3). This similarity implies that the line width cannot be broaden significantly. As a result, the signal line still safely locates within a single LOFAR frequency bin. Furthermore, the effect of inhomogeneities on energy dispersion diminishes quickly as the converted photons rapidly become relativistic after leaving the conversion layer and as the electron density $n_e$ decreases. Therefore, the ray-tracing simulation of radio photon propagation[96] considers angular dispersion due to inhomogeneities while ignoring energy dispersion.

In the context of Eq. (6), the term $P_{sur}$ corresponds to the survival probability of the converted photon. It is important to note that for each converted photon, $P_{sur}$ also depends on the path it travels. Therefore, numerical simulations are essential for accurately calculating $P_{sur}$. The $\beta$ factor in Eq. (6) parameterizes the scattering effect and is defined as

$$\beta(f) = \frac{d^2}{R_S^2} \int_{beam} \frac{g(\theta_1, \phi_1)}{r^2} dS, \tag{7}$$

where $g(\theta_1, \phi_1)$ is the angular distribution function of scattered photons at the last scattering radius $R_S$, beyond which the scattering process can be neglected. The value of $R_S(f)$ is determined by numerical simulation and typically ranges from about 5 to $7R_\odot$, with a slight dependence on photon frequency. The integration in Eq. (7) is over the last scattering surface, and $r$ signifies the distance from the integrated surface element $dS$ to LOFAR. The detailed derivation and computation of Eq. (6) involve intricate but fundamental geometric analyses, and are presented in the methods, subsection The effective spectral flux density received by LOFAR stations.

For simulating the propagation of converted photons within the corona plasma, considering both absorption and scattering effects, we employ the Monte Carlo ray-tracing method developed in ref. 96. We describe the scattering process of radio waves using the Fokker-Planck and Langevin equations based on the Hamilton equations for photons[96–98]. In our simulation, we utilize the Kolmogorov spectrum to describe electron density fluctuations in the quiet Sun, with $\delta n_e / n_e = 0.1$, following the work of ref. 95. Additionally, we consider the anisotropic density fluctuation magnitude as $\alpha_{anis} = 0.1$[95]. Here, $\alpha_{anis}$ represents the anisotropy parameter, which is the ratio between the perpendicular and parallel correlation lengths[96].

Then for each frequency, we calculate $P_{sur}(f)$ and $\beta(f)$, and simulation results are presented in Fig. 2. It is noticeable that the absorption effect becomes more prominent as the frequency increases. Similarly, the smearing effect exhibits a similar trend, primarily due to the diminishing FOV of LOFAR with increasing frequency. This reduction in FOV at higher frequencies amplifies the impact of the smearing effect on the observations.

**LOFAR data analysis and setting constraints on the ultralight DM couplings**

LOFAR is an advanced radio interferometer with high resolution and sensitivity. The observation data of the beam-formed mode[87], where 24 LOFAR core stations in the Netherlands are combined to form 127 tied-array beams. This mode offers significantly increased frequency resolution while reducing spatial resolution. However, the 3.5 km baseline of the LOFAR core limits the FOV to only about 5′ at 32 MHz[87]. The observation data we use is the spectral flux density calibrated in solar flux unit (sfu) within the frequency range of 30-80 MHz. Since some beams are outside the solar surface, only beams with fluxes greater than half of the maximum beam flux are selected. We have data from three different observation periods, all with an observation duration of 17 minutes, which were carried out on 25 April 2015, 3 July 2015, and 3 September 2015. The bandwidth is $B_{res} = 97$ kHz.

The data from the selected beams is averaged. The resulting averaged data is distributed across 516 frequency bins, with each bin

containing 6000 time bins. To eliminate burst-like noises, a data-cleaning process is employed. Firstly, the 6000-time series is divided into 150 intervals, each encompassing 40 time bins, which is sufficient for capturing statistical behavior. The interval with the lowest mean is selected as the reference interval. Subsequently, the mean $\mu_t$ and standard deviation $\sigma_t$ of each interval are compared to those of the reference interval. Intervals meeting the conditions $\mu_t[\text{test}] < \mu_t[\text{ref}] + 2\sigma_t[\text{ref}]$ and $\sigma_t[\text{test}] < 2\sigma_t[\text{ref}]$ are retained. This data-cleaning process only removes transient noises while preserving the time-independent ultralight DM signal.

After data cleaning, for each frequency bin, $i$, we can get the average value $\bar{O}_i$ and the standard deviation $\sigma_{\bar{O}_i}$ as the statistical uncertainty of the time series. We parameterize the background locally by fitting each frequency bin and its adjacent $k$ bins with a polynomial function of degree $n$. In practice, we choose $k = 10$ and $n = 3$. Then, we use the least square method to evaluate the deviation of data to the background fit. The fitting deviation is taken to be the systematic uncertainty $\sigma_i^{sys}$. The total uncertainty is in the quadrature form, $\sigma_i^2 = \sigma_{\bar{O}_i}^2 + (\sigma_i^{sys})^2$. It turns out that $\sigma_i^{sys}$ always dominates in $\sigma^i$.

We adopt the log-likelihood ratio test method[99] to set upper limits on the DPDM parameter space. We construct the likelihood function for a specific frequency bin $i_0$ in the Gaussian form[100]

$$L(S,a) = \prod_{i=i_0-5}^{i_0+5} \frac{1}{\sigma_i\sqrt{2\pi}} \exp\left[-\frac{1}{2}\left(\frac{B(a,f_i) + S\delta_{ii_0} - \bar{O}_i}{\sigma_i}\right)^2\right], \quad (8)$$

where $B(a,f_i)$ is the polynomial function used for background fitting, the coefficients $a = (a_1, a_2, a_3)$ are treated as nuisance parameters, $S$ denotes the assumed DPDM-induced signal at bin $i_0$, and $\delta_{ii_0}$ is the Kronecker delta. We then build the following test statistic[99,100]

$$q_S = \begin{cases} -2\ln\left[\frac{L(S,\tilde{a})}{L(\hat{S},\hat{a})}\right], & \hat{S} \leq S \\ 0, & \hat{S} > S \end{cases}. \quad (9)$$

In the denominator, the likelihood $L$ gets maximized at $a = \hat{a}$ and $S = \hat{S}$; in the numerator, $L$ gets maximized at $a = \tilde{a}$ for a specified $S$. The test statistic $q_S$ follows the half-$\chi^2$ distribution, with the probability density function

$$h(q_S|S) = \frac{1}{2}\delta(q_S) + \frac{1}{2}\frac{1}{\sqrt{2\pi}}\frac{1}{\sqrt{q_S}}e^{-q_S/2}, \quad (10)$$

the cumulative distribution function of which is given by $H(q_S|S) = 1/2\left(1 + \text{erf}\left(\sqrt{q_S/2}\right)\right)$, where $\text{erf}(x)$ is the Gauss error function. Then, we can define the following criterion[99,100]:

$$p_S = \frac{1 - \text{erf}\left(\sqrt{q_S/2}\right)}{1 - \text{erf}\left(\sqrt{q_0/2}\right)}, \quad (11)$$

which measures how far the assumed signal is away from the null result $S = 0$. To obtain the 95% confidence level (C.L.) upper limit $S_{\text{lim}}$, we set $p_S = 0.05$. The results of $S_{\text{lim}}$ as functions of frequency are shown in Fig. 3, with the datasets used stemming from three observation periods represented by different colors. Among the constraints from the three datasets, we select the strongest constraint at each frequency bin to determine the final upper limit. In the 112th frequency bin with $f = 40.6$ MHz for all three periods of observations, an increasing intensity was observed. However, this bin was identified as a bad channel and subsequently excluded from our analysis. Moreover, similar issues were identified in the 25th, 26th, and 27th bins ($f = 32.2$ MHz), the 34th bin ($f = 33.0$ MHz), and the 101st bin ($f = 39.5$ MHz) of the observations on 25 April 2015, in the 46th bin ($f = 34.2$

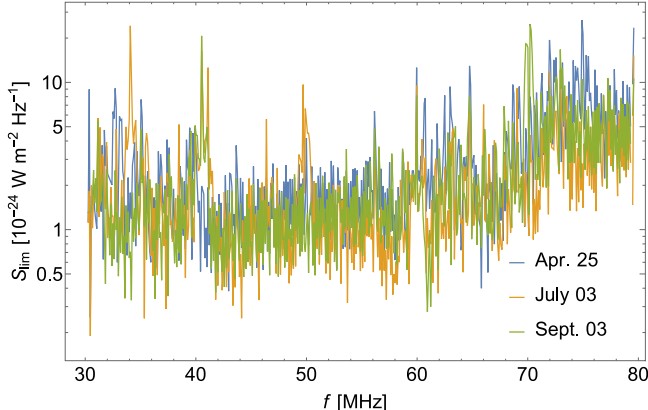

**Fig. 3 | The model-independent constraints on the monochromatic signal.** Model-independent 95% C.L. upper limits $S_{\text{lim}}$ regarding photon frequency $f$ are derived from LOFAR data on a constant monochromatic signal. The limits obtained from the observation data on 25 April 2015, 3 July 2015, and 3 September 2015 are represented by the blue, orange, and green curves, respectively.

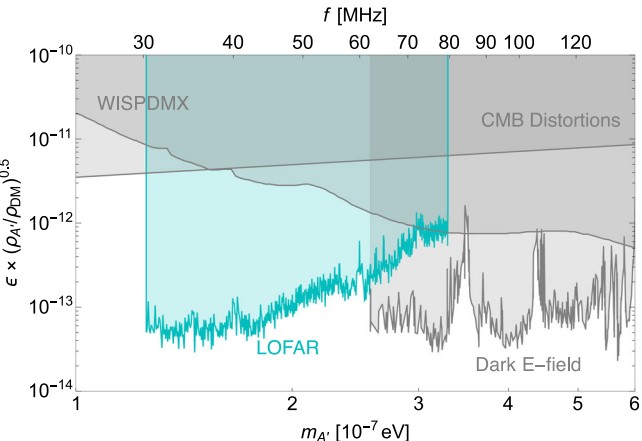

**Fig. 4 | The constraints on the parameter space of DPDM.** 95% C.L. upper limit on the kinetic mixing parameter $\epsilon$ for DPDM regarding the DPDM mass $m_{A'}$ from 17 minutes observation of LOFAR data is shown in the cyan shaded region. We also show the existing constraints (summarized in ref. 52) from the CMB distortion (95%)[6,53], the haloscope searches WISPDMX (95%)[127], and Dark E-field experiment ($5\sigma$)[101] in gray shaded regions. Different constraints may choose different confidence levels, and we keep their original choice unchanged as labeled in the parentheses following each experiment. The existing constraints also assume the dark matter density $\rho_{\text{DM}} = 0.3$ GeV cm$^{-3}$, the same as our choice, and are scaled by the dark photon density $(\rho_{A'}/\rho_{\text{DM}})^{0.5}$.

MHz) and the 208th bin ($f = 50.0$ MHz) of the observations on 3 July 2015. These bad channels were also removed from our analysis.

We calculate the 95% C.L. upper limits on the kinetic mixing parameter $\epsilon$ for the DPDM model by requiring $S_{\text{lim}}$ equal to $S_{\text{sig}}$ in Eq. (6). The upper limit on $\epsilon$ derived from LOFAR data for DPDM is depicted in Fig. 4, which shows that the upper limit on $\epsilon$ is about $10^{-13}$ within the frequency range 30–80 MHz. It is about one order of magnitude better than the existing CMB constraint[6,53], and is complementary to other searches for DPDM at higher frequency, such as the Dark E-field experiment[101].

Based on the same data analysis method, we can set upper limits on the axion-photon coupling for the case of axion DM. However, due to the relatively weak solar coronal magnetic field, our resulting constraint for the axion case is not as strong as many existing constraints. This portion of our analysis is detailed in the methods, subsection Constraint on axion-like particle dark matter.

## Discussion

When DPDM or axion DM pass across the Sun, they can resonantly convert into EM waves in the solar corona. To explore this phenomenon, we conducted numerical simulations of the converted photons propagating in the plasma, including the effects of absorption and scattering. Radio telescopes for solar observations are capable of detecting the monochromatic converted EM waves. We used three datasets of 17-minute observation data from LOFAR to search for such signals. We found that this method sets a stringent limit on the kinetic mixing parameter for dark photons, specifically $\epsilon$ at approximately $10^{-13}$, within the frequency range 30–80 MHz. This limit is about one order of magnitude stronger than the constraint derived from CMB observations. Similarly, we obtain an upper limit on the axion-photon coupling $g_{a\gamma\gamma}$ for the axion DM model in the same frequency range. The constraint on $g_{a\gamma\gamma}$ is better than that from Light-Shining-through-a-Wall experiments but is not comparable with the CAST and astrophysical bounds. The LOFAR data analysis in this work shows great potential in searching for ultralight DM with radio telescopes. With greater sensitivity, we expect future radio programs such as the SKA telescope are expected to yield even greater sensitivity in the search for DPDM and axion DM. Terrestrial radio telescopes cannot search for DPDM with frequencies lower than 10 MHz due to the screening effect from the ionosphere. In these cases, the use of solar probes, such as the STEREO[102] satellite and the Parker Solar Probe[103], equipped with radio spectrometers, could offer an avenue for DPDM detection.

## Methods

### The solar model

In our study, we centered our attention on the quiet Sun due to its reduced occurrence of active events like turbulence and flares. To conduct our calculations, we utilized the electron number density ($n_e$) profile derived from LOFAR observations[89], which employed ray-tracing simulations to fit the solar intensity profile observed by LOFAR in the frequency range of 30–80 MHz.

There have been other density profiles for the quiet Sun, but their differences are within factor of a few. For example, the density profiles based on the work of V. De La Luz, et al.[104], are derived from the temperature ($T$) and hydrogen density ($n_H$) profiles for the quiet Sun, based on the photospheric model from ref. 105 and the coronal model from refs. 106,107. The consistency and validation of these profiles have been confirmed by various research groups[108] using a chromosphere model from ref. 109 and the coronal model from refs. 106,107. These independent calculations consistently agree with each other and have been validated by observations of atomic lines in the soft X-ray range[110] and extreme ultraviolet range[105].

Furthermore, one can adopt a spherically symmetric and hydrostatic model for the quiet Sun, where gas pressure and gravitational force remain in equilibrium, resulting in a static configuration over time. The hydrostatic equilibrium in the quiet Sun region has been confirmed in previous studies[105,110]. Here we provide a simple analytical expression to parameterize the hydrostatic density model, which can be expressed in an exponential form. In this simplified form, the electron density is modeled as[89]

$$n_e = N_0 \exp(1/(H_0 r)), \tag{12}$$

with the parameters $N_0$ and $H_0$, and the latter is defined as

$$H_0 = \frac{k_B T}{0.6 m_p g_\odot} \frac{1}{R_\odot^2}, \tag{13}$$

where $R_\odot$ is the solar radius, $k_B$ is the Boltzmann constant, $g_\odot = 274 \text{ m s}^{-2}$ signifies the gravitational acceleration at the coronal base, and 0.6 times the proton mass $m_p$ gives the average particle mass in the corona[111]. The temperature $T$ corresponds to a scale height

temperature determined by both electron and ion temperatures. There are two parameters to be determined: the density $N_0$ and the temperature $T$. To carry out our calculation, $N_0 = 1.6 \times 10^{11} \text{ m}^{-3}$ and $T = 2 \times 10^6$ K are used from LOFAR observation fit[89].

In Fig. 1, we present a comparison between the profile we adopted from LOFAR observations[89], the profile from ref. 104, the hydrostatic profile modeled by Eq. (12), and the $r^{-2}$ profile. The $r^{-2}$ density profile indicates that a constant solar wind speed has been attained[89]. It can be seen that the solar profile from LOFAR observations closely matches the hydrostatic profile at the high-frequency range, but exhibits a slower decline and transitions into the $r^{-2}$ profile at the low-frequency range. This behavior is expected, as it signifies the shift from subsonic plasma flow in the corona to the supersonic solar wind[89].

As a result, the variation in the electron number density ($n_e$) profile for the quiet Sun across different observations remains within a factor of a few. Although this variation does affect the plasma frequency, it is proportional to the square root of the electron number density, and it only shifts the location of the resonant region. Additionally, the derivative of $n_e$ with respect to radius plays a role in determining the conversion probability, yet its effect is also relatively minor. These uncertainties are small and have a negligible effect on the resulting photon signal.

### The conversion probability of $A' \to \gamma$ and the radiation power

The conversion probability for DPDM to photon can be calculated either by quantum field theory (QFT) as a $1 \to 1$ process, or by solving linearized wave equation[86,90,112]. Here we take linearized wave method as an example, providing formulas for conversion probability and radiation power. It is important to note that in this subsection, our formulas are derived from the solar profile without accounting for small-scale fluctuations. In the following subsection, we will provide estimations of the impact of these small-scale fluctuations.

We can eliminate the kinetic mixing term by performing a rotation of the vector fields in Eq. (1) into interaction basis: $A_\mu \to 1/\sqrt{1-\epsilon^2} A_\mu, A'_\mu \to -\epsilon/\sqrt{1-\epsilon^2} A_\mu + A'_\mu$, where $A_\mu$ represents the vector field of photon. In this basis, the equations of motion become

$$\left[ \frac{\partial^2}{\partial t^2} - \frac{\partial^2}{\partial r^2} + \begin{pmatrix} \omega_p^2 & -\epsilon m_{A'}^2 \\ -\epsilon m_{A'}^2 & m_{A'}^2 \end{pmatrix} \right] \begin{pmatrix} A(r,t) \\ A'(r,t) \end{pmatrix} = 0, \tag{14}$$

which are coupled wave equations.

These second-order coupled equation can be approximated to first-order linearized wave equations using the WKB approximation, as the spatial variation of the plasma frequency occurs on a much larger scale than the wavelength of DPDM. Consequently, we have $\partial_t^2 - \partial_r^2 = -\omega^2 - \partial_r^2 \approx -2k_r(k_r + i\partial_r) - m_{A'}^2 - k_T^2$ under a plane wave solution $A(r,t) = \tilde{A}(r) \exp(-i\omega t + ik_r r)$ with frequency $\omega$ and wavenumber $k = \sqrt{\omega^2 - m_{A'}^2}$, where $k_r$ and $k_T$ is the longitudinal and transverse components of momentum $k$. The resulting first-order linearized wave equation can be expressed as

$$(i\partial_r - H_0 - H_I) \begin{pmatrix} \tilde{A}(r) \\ \tilde{A}'(r) \end{pmatrix} = 0, \tag{15}$$

where

$$H_0 = \frac{1}{2k_r} \begin{pmatrix} \omega_p^2 - m_{A'}^2 - k_T^2 & 0 \\ 0 & -k_T^2 \end{pmatrix},$$

$$H_I = \frac{1}{2k_r} \begin{pmatrix} 0 & -\epsilon m_{A'}^2 \\ -\epsilon m_{A'}^2 & 0. \end{pmatrix} \tag{16}$$

This equation can be solved perturbatively by expanding the time-evolution operator in Dyson series[90]. At the first-order, the conversion probability is given by[112]

$$P_{A' \to \gamma} = \left| \int_{r_0}^{r} dr' \frac{-\epsilon m_{A'}^2}{2k_r} e^{i \int_{r_0}^{r'} dr'' \frac{1}{2k_r} \left[ \omega_p(r'')^2 - m_{A'}^2 \right]} \right|^2. \tag{17}$$

This formula can be further simplified to Eq. (4) by using saddle point approximation

$$\int_{-\infty}^{\infty} dr e^{-f(r)} \approx e^{-f(r_0)} \sqrt{\frac{2\pi}{f''(r_0)}}, \tag{18}$$

where $f'(r_0) = 0$. The thickness of resonant layer is on the order of $10^3$ km for the frequency range under consideration. The WKB approximation and the saddle point approximation can be tested even in the presence with small-scale fluctuations, as we will demonstrate in the next subsection. Additionally, we will numerically show that the value of the conversion probability remains unaffected by small-scale fluctuations in the upcoming subsection.

The radiation power can be obtained from the conversion probability. Taking into account the gravitational focus effect and considering incoming DPDM at infinity moving under the influence of the gravitational potential, the radiation power per solid angle is

$$\frac{d\mathcal{P}}{d\Omega} \approx 2 \frac{1}{4\pi} \rho_{DM} \int d\mathbf{v}_0 f_{DM} v_0 \int_0^b dz 2\pi z P_{A' \to \gamma}(v_r)$$
$$= \rho_{DM} \int d\mathbf{v}_0 f_{DM} P_{A' \to \gamma}(v_0) v(r_c) r_c^2, \tag{19}$$

where $z$ is the impact parameter for DPDM, $b = r_c v(r_c)/v_0$ is the maximum impact parameter for $A'$ to reach the conversion layer at $r = r_c, v(r_c) = \sqrt{v_0^2 + 2G_N M_\odot/r_c}$ is the velocity of DPDM at $r_c$, and the radial direction velocity of DPDM at $r_c$ with different impact parameter is $v_r(z) = \sqrt{2G_N M_\odot/r_c + v_0^2 - v_0^2 z^2/r_c^2}$. The factor of 2 accounts for both incoming and outgoing DPDM as incoming DPDM will be totally reflected.

**Impact of small-scale fluctuations on conversion probability**
In this subsection, we will estimate the influence arising from small-scale inhomogeneities in the plasma by incorporating density fluctuations.

Density fluctuation can lead to three main effects: (1) modifying the magnitude of the $A' \to \gamma$ conversion probability by altering $|\vec{\nabla} \omega_p|$; (2) introducing non-spherical modifications to the conversion surface; and (3) introducing scattering and absorption of the converted photons, resulting in smearing of their velocity directions and a reduction in photon flux. The third effect has been addressed in the Results section when accounting for the propagation effects, utilizing the Monte Carlo ray-tracing simulations.

Regarding the first effect, the inclusion of density fluctuations introduces two opposing influences that modify the conversion probability, $P_{A' \to \gamma}$. On one hand, the derivative of electron density with respective to distance becomes larger due to the fluctuations, leading to a decrease in $P_{A' \to \gamma}$. On the other hand, more resonant points where $\omega_p(r) = m_{A'}$ are introduced, which increases $P_{A' \to \gamma}$. It turns out that these two influences cancel each other out, resulting in $P_{A' \to \gamma}$ with density fluctuations remaining the same as the original value. In addition, the non-spherical effect in the second effect is insignificant. In the following, we will quantify the first and second effects.

In ref. 95, an advanced Monte Carlo simulation technique was employed to address density fluctuations and their impact on photon refraction and scattering during plasma propagation. Their findings suggested that refraction and scattering might be the primary factors contributing to the observed lower brightness temperatures in quiet-Sun radio emissions across different frequencies, deviating from expected values. Hence, we adopt their mathematical framework for describing density fluctuations and incorporate it into our own research.

First and foremost, we emphasize that the density fluctuations in the plasma density are relatively small, with an approximate magnitude of[95]

$$\epsilon_e \equiv \Delta n_e/n_e \approx 10\%, \tag{20}$$

and importantly, this fluctuation fraction remains constant as the radial distance changes[95,113].

The probability distribution of plasma density fluctuations is described by the spatial power spectrum. For the solar corona of the quiet Sun, the spatial power spectrum of density fluctuations can be expressed as[114,115]

$$P(q) = C_N^2 q^{-\alpha}, \quad q_o < q < q_i, \tag{21}$$

where $C_N^2$ is the structural constant, $q$ represents the spatial wavenumber, and $\alpha$ corresponds to the power-law exponent, which is chosen as $\alpha = 11/3$ to reflect the Kolmogorov spectrum.

The scale of density turbulence is defined as $l \equiv 2\pi/q$, with $l_i = 2\pi/q_i$ and $l_o = 2\pi/q_o$ denoting the inner and outer scales of the density turbulence, respectively. It is reasonable to assume that $l_o \approx 10^6 l_i$[95]. Consequently, the steep shape of the Kolmogorov-type spectrum for $P(q)$ indicates that density fluctuations predominantly occur on larger scales.

The inner scale, denoted as $l_i$, can be associated with the ion inertial scale, given by the expression

$$l_i = \frac{684}{\sqrt{n_e/cm^{-3}}} \text{ km}. \tag{22}$$

Consequently, for the plasma layers corresponding to frequencies of 30, 40, 60, and 80 MHz, the respective inner scales $l_i$ are estimated to be 0.2 km, 0.15 km, 0.1 km, and 0.075 km.

The spatial power spectrum can be normalized to the variance of the density fluctuations $\langle \Delta n_e^2 \rangle \equiv (\epsilon_e n_e)^2$ as,

$$\int_{q_o}^{q_i} P(q) 4\pi q^2 dq = \langle \Delta n_e^2 \rangle = (\epsilon_e n_e)^2. \tag{23}$$

Using the Kolmogorov spectrum with $\alpha = 11/3$, it can be determined that

$$C_N^2 = \frac{q_o^{\alpha-3}}{6\pi} \langle \Delta n_e^2 \rangle, \tag{24}$$

$$P(q) = \frac{q_o^{\alpha-3}}{6\pi} q^{-\alpha} \langle \Delta n_e^2 \rangle. \tag{25}$$

The fluctuations can be expressed in the Fourier modes,

$$\Delta n_e(r) = \int_1^{\bar{q}_i} d\bar{q} \, \Delta \tilde{n}_e(q) e^{iqr}. \tag{26}$$

We have rescaled the momentum $\tilde{q} \equiv q/q_o$ for convenience. Subsequently, the fluctuations averaged over the length scale $l_o = 2\pi/q_o$ are

$$
\begin{aligned}
\langle \Delta n_e^2 \rangle &= \frac{1}{l_o} \int dr \int_1^{\tilde{q}_i} d\tilde{q} \int_1^{\tilde{q}_i} d\tilde{q}' \, \Delta \tilde{n}_e(q) \Delta \tilde{n}_e(q') e^{i(q-q')r} \\
&= \int_1^{\tilde{q}_i} d\tilde{q} \, \langle \Delta \tilde{n}_e^2(q) \rangle .
\end{aligned}
\tag{27}
$$

Compared with Eq. (23), we have the fluctuations in the momentum space,

$$
\langle \Delta \tilde{n}_e^2(q) \rangle = P(q) 4\pi q^2 q_o = \frac{2}{3} \tilde{q}^{2-\alpha} \langle \Delta n_e^2 \rangle .
\tag{28}
$$

The averaged derivative of $n_e(r)$ with respect to $r$, in the squared form, is then

$$
\begin{aligned}
\langle (n_e')^2 \rangle &\simeq \langle (\Delta n_e')^2 \rangle \\
&= \frac{1}{l_o} \int dr \int_1^{\tilde{q}_i} d\tilde{q} \int_1^{\tilde{q}_i} d\tilde{q}' \, \Delta \tilde{n}_e(q) \Delta \tilde{n}_e(q') qq' e^{i(q-q')r} \\
&= \int_1^{\tilde{q}_i} d\tilde{q} \, \langle \Delta \tilde{n}_e^2(q) \rangle \cdot q^2 \simeq \frac{2}{3} \langle \Delta n_e^2 \rangle \frac{1}{5-\alpha} q_o^2 \tilde{q}_i^{5-\alpha} .
\end{aligned}
\tag{29}
$$

In the first step, the derivative of the background electron density, $n_{e,\text{bkg}}(r)$, has been omitted due to its relatively small magnitude compared to that of the fluctuations. Similarly, the averaged second derivative of $n_e(r)$ with respect to $r$, in the squared form, is

$$
\begin{aligned}
\langle (n_e'')^2 \rangle &\simeq \langle (\Delta n_e'')^2 \rangle \\
&= \frac{1}{l_o} \int dr \int_1^{\tilde{q}_i} d\tilde{q} \int_1^{\tilde{q}_i} d\tilde{q}' \, \Delta \tilde{n}_e(q) \Delta \tilde{n}_e(q') q^2 q'^2 e^{i(q-q')r} \\
&= \int_1^{\tilde{q}_i} d\tilde{q} \, \langle \Delta \tilde{n}_e^2(q) \rangle \cdot q^4 \simeq \frac{2}{3} \langle \Delta n_e^2 \rangle \frac{1}{7-\alpha} q_o^4 \tilde{q}_i^{7-\alpha} .
\end{aligned}
\tag{30}
$$

Next, we are going to examine whether the WKB approximation and saddle-point approximation are threatened by the inclusion of density fluctuations.

Using Eq. (29), we can estimate the typical length scale of density variations as

$$
\delta l_e = \left| \frac{n_e'}{n_e} \right|^{-1} \simeq \left[ \sqrt{\frac{2}{3}} \left( \frac{1}{5-\alpha} \right)^{\frac{1}{2}} \epsilon_e q_o \tilde{q}_i^{\frac{5-\alpha}{2}} \right]^{-1} \simeq 10^{-3} q_o^{-1},
\tag{31}
$$

which turns out to be much larger than the dark photon wavelength, $\delta l_e k_{A'} \approx 30 \gg 1$. Therefore, the WKB approximation applied in deriving Eq. (4) remains justified even with the density fluctuations included.

Another important length scale is the resonant conversion length, $\delta l_{\text{res}} = \sqrt{2\pi/F''(r_c)}$. It is defined as the length along which the phase factor in Eq. (17), $F(r) \equiv \int dr [\omega_p^2(r) - m_{A'}^2]/(2k_{A'})$, changes by $\pi$. This is the length interval which dominantly contributes to $P_{\gamma \to A'}$. We have

$$
\delta l_{\text{res}} = \sqrt{2\pi \frac{k_{A'}}{\omega_p} \frac{1}{\omega_p'}} \simeq \sqrt{2\pi} v_{\text{DM}} \left( \frac{\delta l_e}{k_{A'}} \right)^{1/2},
\tag{32}
$$

which obviously satisfies $\delta l_{\text{res}} \ll \delta l_e$.

Next, we evaluate the robustness of the saddle-point approximation. The crucial criterion is that the second derivative $F''(r)$ plays a dominant role in the Taylor series of $F(r)$ compared with the higher derivative terms (note that at the resonant point, $F'(r) = 0$). Then we

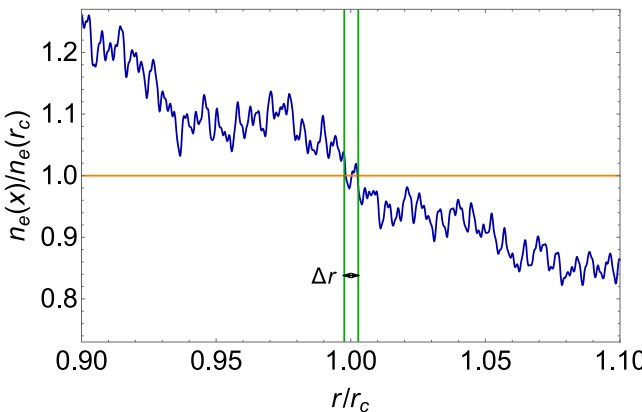

**Fig. 5 | The electron density profile exhibiting fluctuations.** The electron density $n_e$ with fluctuations is shown in blue solid line. This is plotted with the first 12 modes included. This profile is centered around the resonant layer corresponding to 40 MHz. The electron density $n_e(r_c)$ for the 40 MHz frequency is illustrated by the orange line. The two vertical green solid line denotes the interval $\Delta r$ where the intersections can occur.

calculate the following quantity with the help of Eqs. (29) and (30),

$$
\begin{aligned}
\frac{\frac{1}{2!} F''(r)}{\frac{1}{3!} \delta l_{\text{res}} F'''(r)} &\simeq \frac{3}{\sqrt{2\pi}} \frac{[F''(r)]^{3/2}}{F'''(r)} \\
&\simeq \frac{3}{\sqrt{2\pi}} \left( \frac{1}{2k_{A'}} \frac{\omega_p^2}{n_e} \right)^{1/2} \frac{[n'(r)]^{3/2}}{n_e''(r)} \simeq 5.
\end{aligned}
\tag{33}
$$

We see that the second derivative indeed plays a dominant role, indicating that the saddle-point approximation still holds true with an acceptable accuracy. Also, we notice that ref. 116 numerically shows that the saddle-point approximation works well when the ratio in Eq. (33) is larger than unity.

The above arguments show that the form of conversion probability, Eq. (4), is still correct when considering the density fluctuations. However, its numerical value may be altered by the inclusion of density fluctuations. But as we will demonstrate below, the value of $P_{\gamma \to A'}$ remains unchanged. As stated before, there are two new effects that counteract each other in modifying the value of the probability $P_{A' \to \gamma}$: the larger derivative of $n_e$ with respect to distance and more resonant points (>1). An example of an $n_e(r)$ profile with density fluctuations is shown in Fig. 5 where the effects of a larger derivative and more intersections can be seen. We use $r_{\text{dn}}$ to denote the ratio between $P_{A' \to \gamma}$ with and without density fluctuations, and it can be calculated as

$$
\begin{aligned}
r_{\text{dn}} &= \frac{\sum_{n_e(r')=n_e(r_c)} \left| \frac{1}{n_e(r)} \frac{dn_e(r)}{dr} \right|_{r=r'}^{-1}}{\left| \frac{1}{n_{e,\text{bkg}}(r)} \frac{dn_{e,\text{bkg}}(r)}{dr} \right|_{r=r_c}^{-1}} \\
&= \frac{\sum_{n_e(r')=n_e(r_c)} \left| \frac{dr}{dn_e(r)} \right|_{r=r'}}{\left| \frac{dr}{dn_{e,\text{bkg}}(r)} \right|_{r=r_c}} .
\end{aligned}
\tag{34}
$$

We can provide a rough estimate of $r_{\text{dn}}$. Suppose the fluctuation amplitude is $\delta n_e$ in a length scale $\delta r$. The intersections can only occur within the interval $\Delta r$ along which the background density $n_{e,\text{bkg}}(r)$ changes by $\delta n_e$. The number of intersections can be estimated as $\Delta r/\delta r$. Then, we have $r_{\text{dn}} \approx (\Delta r/\delta r) \cdot (\delta r/\delta n_e)/(\Delta r/\delta n_e)$ which is about 1. This suggests that the two effects cancel each other out. Thus, we anticipate that the average conversion probability, when density fluctuations are considered, will remain the same as our original value.

We then proceed to numerically compute the ratio $r_{dn}$ in Eq. (34) using a large sample of $n_e$ profile generated by Monte Carlo method. As we will see below, the concise result $r_{dn} \simeq 1$ is indeed verified.

For the convenience of numerical computation, we first need to discretize the density fluctuations, Eq. (23), as

$$
\begin{aligned}
\langle \Delta n_e^2 \rangle &= \frac{2}{3} \langle \Delta n_e^2 \rangle \int_0^{\log_{10} \tilde{q}_i} \tilde{q}^{3-\alpha} \ln(10) \cdot d\log_{10}\tilde{q} \\
&= \frac{2}{3} \langle \Delta n_e^2 \rangle \sum_{n=0}^{N} \tilde{q}^{3-\alpha} \cdot \Delta \cdot \ln(10) \\
&= \frac{2 \ln(10)}{3} \sum_{n=0}^{N} \langle \Delta n_e^2 \rangle^{(\text{disc.})}(q_n) \cdot \Delta,
\end{aligned}
\tag{35}
$$

where

$$
\begin{aligned}
\langle \Delta n_e^2 \rangle^{(\text{disc.})}(q_n) &\equiv \langle \Delta n_e^2 \rangle \tilde{q}_n^{3-\alpha}, \\
q_n &= 10^{n\Delta} q_o, \quad \tilde{q}_n = 10^{n\Delta}.
\end{aligned}
\tag{36}
$$

$\langle \Delta n_e^2 \rangle^{(\text{disc.})}(q_n)$ is the variance (squared) of density fluctuations of the $q_n$ mode in the interval $[\log_{10}(q_n/q_o), \log_{10}(q_n/q_o) + \Delta]$. The discretization is carried out in the logarithmic scale, as the momentum span is broad, spanning 6 orders of magnitude from $q_o$ to $q_i$. The total number of modes is $N = \log_{10}(q_i/q_o)/\Delta$. Next, the variance of density fluctuations for different momentum modes can be estimated as

$$
\sigma_{n_e}(q_n) \simeq \sqrt{\langle \Delta n_e^2 \rangle^{(\text{disc.})}(q_n)} \simeq \epsilon_e n_e \left( \frac{q_n}{q_o} \right)^{\frac{3-\alpha}{2}}.
\tag{37}
$$

The density as a function of distance, $n_e(r)$, with the fluctuations taken into consideration, can be modeled as

$$
n_e(r) = \frac{r_c^2}{r^2} \left( n_{e,\text{bkg}}(r_c) + \sum_{n=0}^{N} \delta n_e(q_n)\Delta \cdot \sin[q_n(r - r_c) + \phi_n] \right).
\tag{38}
$$

Note that we have used sine functions for simplicity in numerical evaluation. We also add random phases $\phi_n$ for each mode. Based on (37), we have the variance of $\delta n_e(k_n)/n_e(r_c)$,

$$
\frac{\sigma_{n_e}(k_n)}{n_e(r_c)} \simeq \epsilon_e \cdot 10^{n\Delta \cdot (\frac{3-\alpha}{2})}.
\tag{39}
$$

Then, we employ Monte Carlo method to generate the values for $\delta n_e(k_n)$ following a Gaussian distribution with a mean value of zero and a variance of $\sigma_{n_e}(k_n)$ for each $k$ mode. Additionally, the phase $\phi_n$ randomly picks up a value between 0 and $2\pi$ for each $k$ mode. To check the effect of the fluctuations on the conversion probability, we take the frequency 40 MHz as an example. At this frequency, the solar wind dominates so that the background density profile can be taken as $r^{-2}$ as shown in Eq. (38). In this example, the corresponding resonant conversion layer is at $r_c \simeq 9.4 \times 10^5$ km from the solar center, which corresponds to $q_0 r_c \simeq 40$.

To proceed, we take $\Delta = 0.2$ and thus we have $N = 30$ modes in total. However, computing $r_{dn}$ in the presence of the full $N = 30$ modes turns out to be challenging due to numerical limitations. Consequently, we perform the calculations for subsets of modes, specifically considering the first $2, 4, 6, \ldots, 20$ modes individually. For each chosen number of modes, we iteratively evaluate $r_{dn}$ 1000 times using different Monte Carlo $n_e$ profiles and then take the average to ensure statistical stability. In Fig. 6, we present the result of $r_{dn}$ with more modes gradually included in computations. We see that the average value of $r_{dn}$ converges to approximately 1, insensitive to the number of modes. Therefore, we conclude that including density fluctuations does not significantly change the value of $P_{A' \to \gamma}$, as the two effects of larger derivatives and more resonant points cancel out each other.

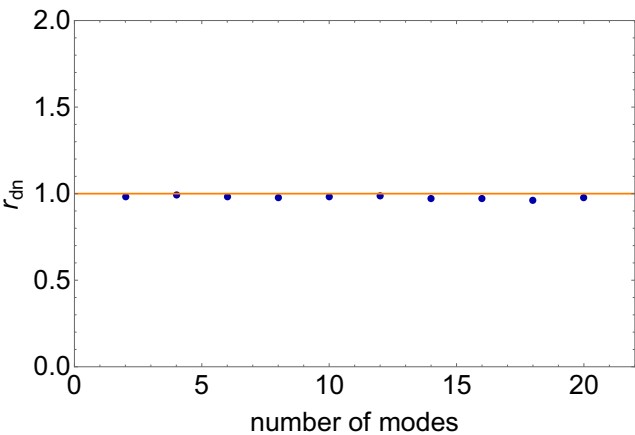

**Fig. 6 | The ratio between the conversion probabilities with and without density fluctuations.** The ratio $r_{dn}$ is computed numerically for various numbers of modes $k$ at the 40 MHz frequency, and is shown as the blue dots, while the orange line marks the position of unity as a reference. These calculations are performed over a total of 1000 samples and the resulting values are averaged.

Next, we check the second effect of density fluctuations, which concerns the modification to the shape of the conversion surface. The deformation of the conversion surface is within the length interval $\Delta r$ around $r_c$. As illustrated in Fig. 6, $\Delta r/r_c \ll 1$. Therefore, the deformation effect is negligible compared with the orignal conversion sphere located at $r = r_c$ without density fluctuations included.

In summary, we have provided a quantitative demonstration that inhomogeneities have a minimal impact on our calculations. This conclusion holds true for the condition of deriving the conversion probability, the magnitude of the conversion probability, and the deformation of the conversion sphere. There are two key factors contribute to the result: Firstly, the fraction of density fluctuation remains small, at approximately 10%. Secondly, the density fluctuation predominantly occurs at larger scales, indicating that small-scale turbulence has a limited effect.

### The effective spectral flux density received by LOFAR stations

Firstly, the Field of View (FOV) of LOFAR, or effectively, the Full Width Half Maximum (FWHM) of LOFAR, is determined by

$$
\text{FWHM} = \eta \times \frac{\lambda}{D},
\tag{40}
$$

where $\lambda$ is the observation wavelength, the coefficient $\eta = 1.02$[117], and $D \simeq 3.5$ km is the station diameter according to ref. [118]. Therefore, the FWHM (for one beam) is approximately $10^{-3}$ rad.

We can effectively define the last scattering sphere of radius $R_S$, beyond which the scattering effect can be ignored, allowing the radio waves to propagate in straight lines for $r > R_S$. The total radiation power for dark photon signal at frequency $f$ after conversion is $d\mathcal{P}/d\Omega \times 4\pi$. Therefore, the survived power at the last scattering sphere is given by

$$
\mathcal{P} = P_{\text{sur}}(f) 4\pi \frac{d\mathcal{P}}{d\Omega}.
\tag{41}
$$

Considering a virtual source point $P_1$ situated within a surface element $dA_1$ on the last scattering sphere (as depicted in the schematic diagram of Fig. 7), the power it radiates in the direction $\mathbf{r}$ is

$$
d\mathcal{P}' = \mathcal{P} \frac{dA_1}{4\pi R_S^2} g(\theta_1, \phi_1) d\Omega_1,
\tag{42}
$$

where the angular distribution function $g(\theta_1, \phi_1)$ accounts for the fact that after multiple random scattering events, the radiation from the

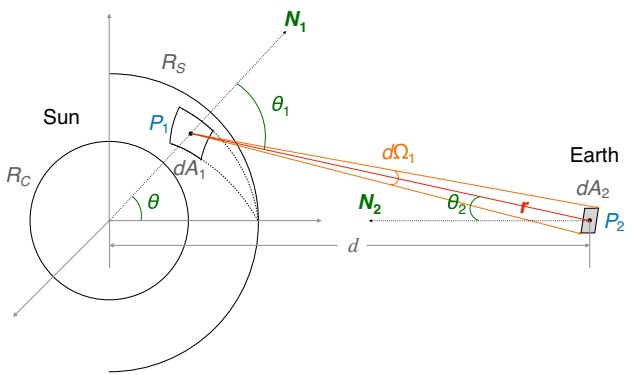

**Fig. 7 | Schematic diagram of the propagation of photons after the last scattering.** $R_C$ denotes the conversion layer, and $R_S$ denotes the last scattering sphere. A surface element $dA_1$, which containing a point $P_1$, acts as the radiation source on the last scattering sphere. $\theta$ is the polar angle of $P_1$. Another surface element $dA_2$, encompassing $P_2$, serves as the detection area on the Earth, which defines a solid angle $d\Omega_1$ about $P_1$ in the direction of **r**. $\theta_i$ is the angle between the propagation vector **r** and the normal vector $\mathbf{N_i}$ of $dA_i$. The direction of $\mathbf{N_2}$ is aligned with the line connecting the centers of the Sun and the Earth. $d$ is the distance from the Earth to the Sun.

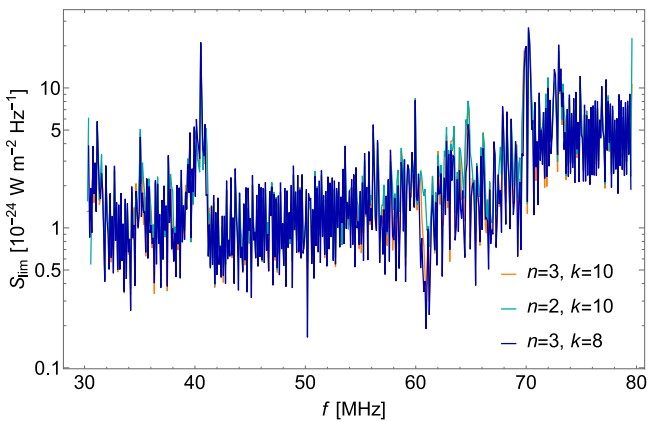

**Fig. 8 | The constraints on the monochromatic signal with different background fitting parameters.** The 95% C.L. upper limits from LOFAR data on September 3, 2015 with a constant mono-chromatic signal using different background fitting parameters. The orange, cyan and blue limits represent using 10 adjacent bins with a 3rd-degree polynomial, 10 adjacent bins with a 2nd-degree polynomial and 8 adjacent bins with a 3rd-degree polynomial, respectively, with n representing the degree of polynomial and k representing the number of adjacent bins.

surface element is not simply in the radial direction. $g(\theta_1, \phi_1)$ is normalized as

$$1 = \int g(\theta_1, \phi_1) d\Omega_1. \tag{43}$$

The relation $d\Omega_1 = dA_2 \cos\theta_2 / r^2$ is useful where the cosine factor accounts for converting the receiving area $dA_2$ to the projected area normal to **r**. Then, Eq. (42) becomes

$$d\mathcal{P}' = \mathcal{P} \frac{dA_1}{4\pi R_S^2} g(\theta_1, \phi_1) \frac{dA_2 \cos\theta_2}{r^2}, \tag{44}$$

where $r$ is the distance from the surface element to the Earth. Meanwhile, since $\theta_2$ is on the order of $10^{-3}$ rad, it follows that $\cos\theta_2 \simeq 1$.

By substituting Eq. (41) into Eq. (44) and integrating over the area on the last scattering sphere covered by the beams, the effective spectral flux density (power per unit area and unit frequency) received

by LOFAR is derived as:

$$S_{\text{sig}} = P_{\text{sur}} \frac{1}{\mathcal{B}} \frac{1}{R_S^2} \frac{d\mathcal{P}}{d\Omega} \int_{\text{beam}} \frac{g(\theta_1, \phi_1)}{r^2} dA_1. \tag{45}$$

As discussed in the main text, the angular distribution function $g(\theta_1, \phi_1)$ can be determined by numerical simulations. The integration is performed in the spherical coordinates $(\theta, \phi)$ with the Solar center as the origin. Consequently, it can be transformed into

$$S_{\text{sig}} = P_{\text{sur}} \frac{1}{d^2} \frac{1}{\mathcal{B}} \frac{d\mathcal{P}}{d\Omega} \int_{\text{beam}} g(\theta_1, \phi_1) \frac{\sin\theta_1}{\cos\theta_2} d\theta_1 d\phi_1. \tag{46}$$

where $d = 1$ AU is the distance from the Earth to the Sun. $\cos\theta_2 = \sqrt{1 - R_S^2 \sin\theta_1^2 / d^2}$ is the geometric relation. The $R_S$ dependence in $\cos\theta_2$ is canceled out by the implicit $R_S$ dependence in $g(\theta_1, \phi_1, R_S)$. For the simplest scenario without scattering, $g(\theta_1, \phi_1) = \delta(\theta_1) / (2\pi \sin\theta_1)$, Eq. (46) becomes $S_{\text{sig}} = P_{\text{sur}} \cdot 1/d^2 \cdot 1/\mathcal{B} \cdot d\mathcal{P}/d\Omega$, as expected. It is worth noting that since the data is averaged over the beams with flux larger than 50% of the maximum beam flux, the spherical surface integral is over the area covered by these selected beams, and then divided by the number of selected beams.

## Statistics of robustness of background fitting parameter choosing

The upper limits on mono-chromatic signals, determined through the log-likelihood ratio test, exhibit robustness against variations in the parameters used for fitting the background. These parameters, denoted as $n$ for the degree of the polynomial function and $k$ for the number of bins included in the calculation, do not significantly affect the results. Typically, quadratic and trilinear polynomial forms are employed, yet even with these different choices, the outcomes remain largely unchanged. To demonstrate this resilience, we conducted a comprehensive analysis on LOFAR data collected on September 3, 2015, using varying degrees of polynomials and numbers of adjacent bins. Specifically, we examined three cases: 10 adjacent bins with a 3rd-degree polynomial, 10 adjacent bins with a 2nd-degree polynomial, and 8 adjacent bins with a 3rd-degree polynomial. The results of these analyses are presented in Fig. 8.

Our investigation demonstrates that the derived signal limits exhibit a remarkably stability and are impervious to the specific choices of $n$ and $k$. This robustness serves to reinforce the reliability and consistency of our method in establishing upper limits on the mono-chromatic signal from the LOFAR data.

## Statistics of the Gaussian feature of LOFAR data

In our fitting process, the flux $F(t_i, f_j)$ is characterized by its time index $t_i$ and frequency index $f_j$. To analyze each frequency bin $f_j$, we calculate the average flux over time, denoted as $\bar{F}(f_j)$, and assume that it varies smoothly in frequency, which is fitted by using 3rd-degree polynomials. Within a fixed frequency bin, we consider the fluxes of different time bins to follow a Gaussian distribution, with $\bar{F}(f_j)$ serving as the mean of the Gaussian function.

To validate the assumption of Gaussian distribution, we specifically examine two frequency bins ($j = 200, 400$), corresponding to 49.21 MHz and 68.74 MHz on 3 September 2015, respectively. After undergoing the data cleaning process, each bin contains 920 and 1040 time bins, respectively. The top and bottom panels of Fig. 9 display the histograms for the flux at $f_{200}$ and $f_{400}$, respectively, and these plots align well with the Gaussian distribution.

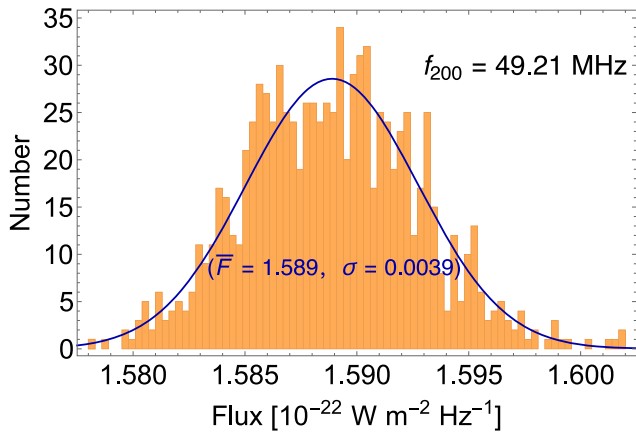

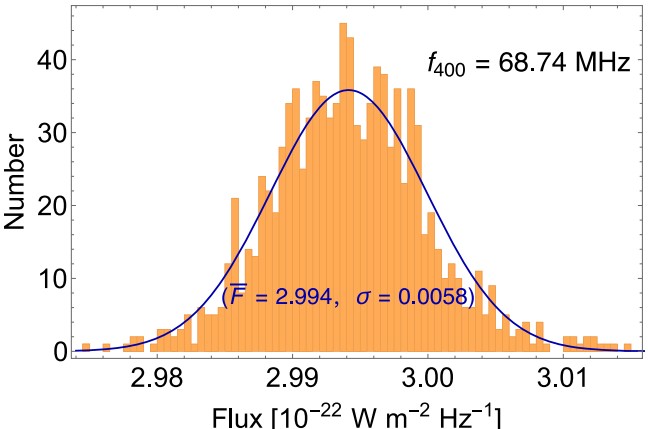

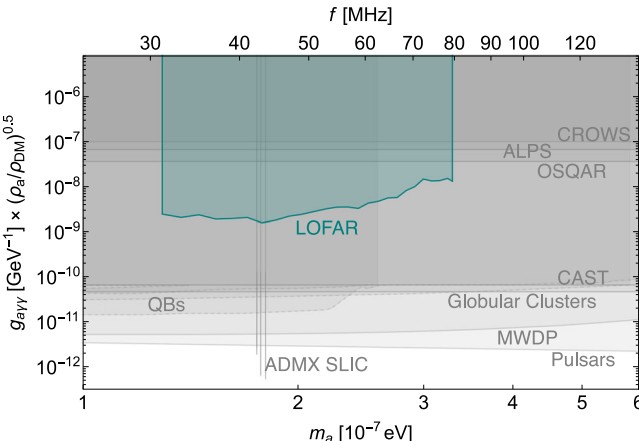

**Fig. 9 | The distributions of flux in the LOFAR data.** They are obtained from time-bins after data cleaning process on September 3, 2015. **a** The distribution for the 200th bin with $f_{200} = 49.21$ MHz is shown in the orange shaded region, while the Gaussian distribution with mean value $\bar{F} = 1.589$, standard deviation $\sigma = 0.0039$ is shown in the solid blue line. **b** The distribution for the 400th bin with $f_{400} = 68.74$ MHz is shown in the orange shaded region, while the Gaussian distribution with mean value $\bar{F} = 2.994$, standard deviation $\sigma = 0.0058$ is shown in the solid blue line. These distributions exhibit a good fit to a Gaussian distribution.

**Fig. 10 | The constraints on the parameter space of axion-like particle dark matter.** 95% C.L. upper limit on axion-photon coupling $g_{a\gamma\gamma}$ from 17 minutes observation of LOFAR data is shown in the cyan shaded region. We also show the existing constraints (summarized in ref. 52) from various experiments and astrophysical observations in gray shaded regions, including Light-Shining-through-a-Wall experiments: CROWS[122] (95%), ALPS[123] (95%), and OSQAR (95%)[124]; helioscope: CAST (95%)[125]; haloscope: ADMX SLIC (90%)[126]; astrophysical bounds: magnetic white dwarf polarization (MWDP) (95%)[70], Globular Clusters (95%)[82,83], pulsars (95%)[62], as well as quasars and blazars (QBs, shown in dashed gray) (95%)[75–77]. Different constraints may choose different confidence levels, and we keep their original choice unchanged as labeled in the parentheses following each experiment. The ADMX SLIC constraint assumes axions to be dark matter, $\rho = 0.45$ GeV cm$^{-3}$, and we have rescaled it to be 0.3 GeV cm$^{-3}$ in the plot for comparison.

## Constraint on axion-like particle dark matter

In the axion DM model, the axion $a$, as a pseudo-scalar particle, interacts with the SM photon via

$$\mathcal{L}_{a\gamma} = \frac{1}{2}\partial_\mu a \partial^\mu a - \frac{1}{2}m_a^2 a^2 + \frac{1}{4}g_{a\gamma\gamma} a F_{\mu\nu}\tilde{F}^{\mu\nu}, \tag{47}$$

where $\tilde{F}^{\mu\nu} \equiv \varepsilon^{\mu\nu\alpha\beta}F_{\alpha\beta}/2$ is the dual EM field strength, $m_a$ is the axion mass, $a$ is the axion field and $g_{a\gamma\gamma}$ is the coupling strength between the axion and EM field. The last term in (47) can be simplified as $-g_{a\gamma\gamma}a\mathbf{E}\cdot\mathbf{B}$.

Similar to the dark photon scenario, the probability of axion DM converting into photons is given by

$$P_{a\to\gamma}(v_{rc}) = \pi \frac{g_{a\gamma\gamma}^2 |\mathbf{B}_T|^2}{m_a} v_{rc}^{-1} \left| \frac{\partial \ln \omega_p^2(r)}{\partial r} \right|_{r=r_c}^{-1}. \tag{48}$$

$\mathbf{B}_T$ is the magnetic field transverse to the direction of the axion propagation. The key difference from the dark photon case is that, the conversion of axions into photons requires the presence of a magnetic field. The probabilities (4) and (48) in the two cases are related via the

expression

$$\sqrt{\frac{2}{3}}\epsilon m_{A'}^2 \Longleftrightarrow g_{a\gamma\gamma}|\mathbf{B}_T|m_a. \tag{49}$$

The Sun possesses a dipole-like magnetic field but suffers from large fluctuations[119,120]. The global map of the magnetic field in solar corona obtained using the technique of the Coronal Multi-channel Polarimeter shows that the magnetic field strength is about 1-4 Gauss in the corona at the distance of 1.05-1.35 $R_\odot$[121]. In our case, the resonant conversion happens at the range of about 2.18-1.12$R_\odot$ (corresponding to frequencies in the range of 30–80 MHz; see Fig. 1). To proceed conservatively, we estimate $|\mathbf{B}_T|$ to be 1 Gauss at 1.05$R_\odot$ and extrapolate this value to obtain $|\mathbf{B}_T| \approx 0.11$-0.82 Gauss for our frequency range, following the attenuation relation $\propto R^{-3}$.

The upper limit for the dark photon case can be directly translated into that for the axion case using the relation (49). We adopt $|\mathbf{B}_T|$ as a function of distance using the extrapolation above. Consequently, we plot the constraint on $g_{a\gamma\gamma}$ in Fig. 10. However, there is a large uncertainty in our estimation of the magnetic field, which overshadows other statistical and systematic uncertainties. Therefore, the zig-zag features shown in Fig. 4 become less meaningful in the axion DM case. As a result, in Fig. 10, we average the upper limits over every 20 frequency bins to indicate the sensitivity of the LOFAR data on axion DM model. The resulting graph shows that while our limit exceeds the existing constraints from Light-Shining-through-a-Wall experiments, including CROWS[122], ALPS[123], and OSQAR[124], it is not as competitive as the direct search experiments such as CAST[125] or ADMX SLIC[126] (in very narrow bands), and the astrophysical bounds from observations of magnetic white dwarf polarization[70], Globular Clusters[82,83], pulsars[62], as well as quasars and blazars[75–77].

## Data availability

The LOFAR data used in this work is available at https://github.com/Link23GH/UDM_LOFAR. The data that support the plots and findings of this work are provided as a source data file. Source data are provided with this paper. The datasets generated during and/or analyzed during the current study are available from the corresponding authors upon request. Source data are provided with this paper.

## Code availability

The codes that support the plots and findings of this work are available from the corresponding authors upon request.

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

## Acknowledgements

The authors would like to thank Li Feng, Zongjun Ning, Baolin Tan, Chengming Tan, and Qiang Yuan for helpful discussions. The authors would like to express a special thanks to Eduard Kontar for helpful discussions and especially the interpretation of the data format and calibrations. The work of HA is supported in part by the National Key R&D Program of China under Grants No. 2021YFC2203100 and No. 2017YFA0402204, the NSFC under Grant No. 11975134, and the Tsinghua University Dushi Program No. 53120200422. The work of SG is supported by NSFC under Grant No. 12247147, the International Postdoctoral Exchange Fellowship Program, and the Boya Postdoctoral Fellowship of Peking University. The work of JL is supported by NSFC under Grant No. 12075005, 12235001, and by Peking University under startup Grant No. 7101502458.

## Author contributions

The authors are listed in alphabetical order. H.A. and J.L. initiated and supervised the work; S.G. and Y.L. developed the method; Y.L. did the ray-tracing simulation, the geometric calculation, and the data analysis, with substantial contributions from S.G; S.G. and Y.L. wrote the initial manuscript with editions from H.A. and J.L; X.C. provided expertize on LOFAR observations and the ray-tracing code.

## Competing interests

The authors declare no competing interests.
