## [Peer Review File · Nature Communications]

REVIEWER COMMENTS

Reviewer #1 (Remarks to the Author):

The manuscript "Searching for Ultralight Dark Matter Conversion in Solar Corona using LOFAR Data" uses observations by the LOFAR observatory of the Sun to look for the resonant conversion of ambient dark matter in the solar corona. The authors focus on two dark matter candidates, the dark photon and the axion, both with masses between $\sim 1.5e-7$ and $\sim 3e-7$ eV. The constraints on the dark photon are the strongest to date across most of the mass range, while those on the axion are extremely sub-dominant to an array of other laboratory and indirect axion searches.

Light dark photons have become an increasingly studied dark matter candidate over the past few years. For this reason, I believe that the strong constraints derived on the dark photon mixing are sufficient to consider this manuscript for publication. The authors attempt to make conservative assumptions and use up-to-date data and analysis tools, providing confidence that the constraints are not being dramatically oversold. Having said that, I do have a few concerns that I would like the authors to address before I can recommend this publication. These include:

1.) At the moment the authors have treated the entire photon production process as being one-dimensional (i.e. they assume in writing equations 4, 5 and 7 that all dark photons travel on radial trajectories). This is clearly not a correct assumption. Equation 7 should contain an angular factor which arises from the fact that the flux of dark photons across a surface depends on the orientation of the velocity and the surface normal. Moreover, the conversion probability changes as one considers non-radial trajectories. The latter can potentially also be quite sensitive to small scale inhomogeneities in the plasma, should they exist (in other words, this raises the question: can the conversion surface truly be treated as spherical?). I think a more careful treatment is required.

2.) The authors do not address what the width of the line should. Presumably this means they are neglecting broadening effects? It is not clear to me whether this is valid. In general, this will depend on the velocity of the plasma, the scale and shapes of the inhomogeneities, and the number of scatterings (not to mention the broadening induced as the photon accelerates away from the production point). The authors could incorporate this direction into their ray tracing procedure -- should they do this, the authors should also be careful to show how the assumptions currently adopted on the scattering surface alter this result. I believe the authors need to more firmly establish that the line will end up in a single bin.

3.) I have a number of concerns regarding the authors statistical treatment. First, why have the authors chosen to use $k=10$ and $n=3$? What is the impact of changing these to other reasonable values? Can the authors show samples of how well these fits work, and what the residuals look like? Are the residuals Gaussian? If not, then it doesn't make sense to adopt the Gaussian approximation. Its also unclear to me whether it is really fair to adopt the half- χ^2 distribution -- this is not always a valid assumption, especially if one is dominated by large non-Gaussian systematics. The authors should provide many more details and checks to show that their procedure does in fact reflect the stated sensitivity.

Finally, I have two smaller comments:

1.) I find the results on the axion rather uninteresting -- this is not even remotely competitive, and I suppose that the authors knew this before preceding in this direction. I personally find that including axions tends to distract from the most important message, and I would consider putting it in an appendix, or at the very least de-emphasising the result (leaving the focus on the dark photon).

2.) At the moment the text does not read remarkably well. Articles are frequently missing and sentences seem to be put together in a bit haphazard manner. I would request that the authors dedicate time into improving this.

Reviewer #2 (Remarks to the Author):

An et al 2023

Searching for Ultralight Dark Matter Conversion in Solar Corona using LOFAR Data.

The paper is about setting bounds on whether Dark Photon Dark Matter is observable in solar radio data observed by LOFAR between 30-80 MHz. The authors analyse three periods of solar LOFAR data and perform a confidence level analysis to place upper limits on the detectability of dark matter in the signal, using a log-likelihood ratio test. Each period of solar data is cleaned to remove burst-like noises. The authors conclude a lack of detectable signal in the radio data.

The paper deals with the solar LOFAR data in a mostly robust manor for the analysis. The conclusions appear to be a significant leap over the previous bounds at lower values of m_a . There are a number of considerations that the authors should address before the paper can be accepted.

1. The use of acronyms that are not defined is rife throughout the paper (e.g. WIMP, CP, QCD, SM, CMB, DPDM). All acronyms should be defined the first time they are used.

2. There are also a number of symbols used in the equations and the text that are not defined (e.g. m_A , A , μ , m_{DM} ...). All symbols should be defined the first time they are used.

3. The formula for plasma frequency is incorrectly in units of eV. There is an extra $1/(hc)$ present in the formula for angular plasma frequency.

4. The model that has been used for the quiet Sun is based upon millimeter and submillimeter wavelengths (GHz frequencies), whereas the wavelengths used for the LOFAR analysis are between 1-10 m. The authors should use a standard solar density/frequency model for the Sun that has been derived from relevant wavelengths to the LOFAR observations.

5. The value of the magnetic field chosen is too high for the solar corona at the heights that correspond to LOFAR frequencies. These heights are typically 1.5 to 2 solar radii in altitude, whereas the estimates of magnetic field are taken between 1.05-1.35 solar radii. In addition, the magnetic field of the Sun has a strong dependence with distance, dropping off at or faster than the $1/r^3$ dependence one gets in a standard dipole. The authors should consider this magnetic field dependence, given the importance of B in the calculation of axion conversion.

6. The most probable velocity of 235 km/s at the conversion layer seems oddly specific. Some more information should be given about how/why this velocity is chosen.

7. When talking about the scattering effect, at line 142, it should be made clear that the authors are discussing the tied-array beam mode that was used for the observations in question. Standard interferometric LOFAR observations do not have multiple beams that resolve the source.

8. Line 195 – this statement is half-true. The frequency resolution for beam-formed imaging is increased but the spatial resolution is significantly decreased.

9. Figure 2: it is not clear what the increase in intensity that occurs at 40 MHz comes from. The increase looks like potentially bad channels within the LOFAR data. This conclusion is based upon the triple peak that has occurred whenever there is an increase in the flux, and the choice of $n=3$ (line 227). The authors should look over the data and double check the raw LOFAR data before the sub-bands are created from the individual channels. The intensity does not look like solar radio bursts, as indicated from line 240. These should have been filtered out via the data cleaning, and are typically broader in bandwidth.

10. Figure 4: The authors should discuss the changes when the magnetic field changes with distance. This appears to reduce the gradient of the line, or even reverse the sign depending upon how $B(r)$ is assumed.

Reviewer #3 (Remarks to the Author):

In this work the authors use radio data by the LOFAR telescope to put strong constraints on Dark Photon (DP) and axion dark matter looking at the Sun. The mechanism is the following: when a DP (axion) with mass $m_{A'}$ (m_a) passes through the solar corona, it can resonantly convert into electromagnetic waves if the particle bare mass matches the plasma frequency of the plasma, $m_{A'} = \omega_{\text{pl}}$.

Before passing to some technical comments and questions, let me stress that the paper is easy to read, clear and of great interest for all the particle physics and astrophysics community. In particular, the DP region the authors explore is one of the most motivated one in typical scenarios of physics beyond the standard model, as DPs may easily get produced as dark matter during inflation. Moreover, a large experimental effort exists to probe both DPs and axions dark matter in this mass window, making this work of great interest also for the experimental community.

All in all, I consider the paper of high appeal, and in principle worth to be published in Nature Communication, but only if the authors can address the following comments/questions and make their results more robust.

1) My main concern is about the treatment of the plasma frequency (therefore the electron density) in the plasma corona. The authors assume a "static and spherical" solar model, which seems a too rough approximation to me. First of all, although I understand the adopted profile was reported in previous publications (in PRL), it would be useful for the reader to have it in the Nature publication too. I would then ask the authors to write a small Appendix with the simplified solar model they use, maybe also providing an analytical expression for the electron density, which can be used to reproduce the gross features of their results. Second of all, and most important, I would expect the plasma frequency to have strong temporal variations. In fact, the authors for example recognize that the magnetic field "shows strong magnetic field in-homogeneity depending pending on location, altitude, and time". If the magnetic field varies a lot, I would expect also the electron density to vary consistently. If this is not the case, the authors should justify it with data. In any case, I would ask the authors to look for data about the electron density in the solar corona \textit{during} the 17 minutes of LOFAR data taking.

2) My second concern is connected to the first one, and it is related to the "poor" treatment of uncertainties for the plasma profile in general. Apart from what I wrote above, I also felt that the authors give the idea that the electron density profile in the solar corona is very well known. This should be better justified if so. For example, I would find useful a small Appendix in which the authors explain how the electron density is measured (for example using Fe XIII lines) and how precisely.

3) I would like the authors to show to me how the transmission coefficient changes when $\delta n_e/n_e$ increases. In particular: how much of the signal gets transmitted if (let's say) $\delta n_e/n_e = 0.2 - 0.5$ instead of 0.1? I would expect a sensitive dependence on this quantity, because waves can get more easily reflected back if they can encounter large up-ward fluctuations when they propagate away from the resonant point.

4) The treatment of the background in the likelihood analysis seems a bit arbitrary and no discussion is given to show how sensitive the results are to the choice of a simple polynomial function with $n = 3$. The authors should provide other examples and show how their final plots change accordingly. For example, they can consider a combination of an exponential sine squared kernel (to capture highly oscillatory features in the data) and an exponential-squared kernel (to avoid mismodeling features which do not have strong periodicity). Moreover they should also address the question on how results change varying the considered number of adjacent bins.

5) The fact that the TS follows a half- χ^2 distribution is an assumption, as far as I can see. It should then be validated through MC simulations.

6) For the axion case, I have some concern about the conversion probability. Do the authors know if the magnetic field in the solar corona has a strong turbulent component and/or large variations on small length scales? If this is the case, I am not sure the Landau-Zener approximation, usually adopted to derive the conversion probability, holds in this case.

Dear editors and referees,

We are grateful for your valuable comments and suggestions, and we highly appreciate your positive evaluation of our study. We carefully considered your suggestions and have made revisions to the manuscript accordingly. In the following, we address the specific comments raised in the three referee reports.

Replies to Referee 1

1.1a). *At the moment the authors have treated the entire photon production process as being one-dimensional (i.e. they assume in writing equations 4, 5 and 7 that all dark photons travel on radial trajectories). This is clearly not a correct assumption. Equation 7 should contain an angular factor which arises from the fact that the flux of dark photons across a surface depends on the orientation of the velocity and the surface normal.*

Reply. We thank the referee for pointing this out. Indeed, in our calculation, we have already considered dark photons from all directions. The detailed derivation of Eq. (4) in the manuscript can be found in the supplemental material of [1]. The resonant conversion probability can be written as

$$P_{A' \rightarrow \gamma} = \frac{2}{3} \times \pi \epsilon^2 m_{A'} \left| \frac{\partial \ln \omega_p^2}{\partial t} \right|_{r=r_c}^{-1}, \quad (1)$$

where the time derivative of ω_p^2 is caused by the motion of DPDM. Then we can replace the time derivative of the plasma frequency with the expression:

$$\frac{\partial \omega_p}{\partial t} = |\vec{v} \cdot \vec{\nabla} \omega_p|. \quad (2)$$

By adopting the spherical symmetric and hydrostatic solar model, we assume that the plasma frequency ω_p solely depends on the radius r , denoted as $\omega_p(r)$. With this assumption, we can express Eq. (1) as Eq. (4) in our manuscript:

$$P_{A' \rightarrow \gamma}(v_r) = \frac{2}{3} \times \pi \epsilon^2 m_{A'} v_{rc}^{-1} \left| \frac{\partial \ln \omega_p^2(r)}{\partial r} \right|_{r=r_c}^{-1}, \quad (3)$$

where v_{rc} is the radial velocity at the resonant layer. Furthermore, in Eq. (7) of the manuscript, we have taken the average of all dark photon velocities based on the distribution of the Standard Halo Model.

The reason that we assume the plasma distribution is isotropic is the following. Our study focused on the quiet Sun as it exhibits fewer active events such as turbulence and flares. We adopted a spherically symmetric and hydrostatic model for the quiet Sun, where the gas pressure is in equilibrium with gravitational force, remaining static over time. The hydrostatic equilibrium of the quiet Sun region has been confirmed in previous studies [2, 3]. To conduct our calculations, we considered the electron number density (n_e) and temperature (T) profiles from Ref. [4], which derived the temperature (T) and hydrogen density (n_H) profiles for the quiet Sun based on the photospheric model from Ref. [2] and the coronal model from Refs. [5, 6]. The profiles were calculated by various research groups [7] using a chromosphere model from Ref. [8] and the coronal model from Refs. [5, 6]). These independent calculations consistently agree with each other and have been validated by observations of atomic lines in the soft X-ray range [3] and extreme ultraviolet range [2]. Thus, it demonstrates the simplicity and reliability of the spherical symmetric and hydrostatic solar model employed in this paper.

In summary, we have taken into account the dependence of the dark photon flux on the relative orientation between velocity and surface normal. The adoption of the spherically symmetric and hydrostatic solar model offers a simple and dependable representation of the quiet Sun, with its predictions aligning well with observational data. The movement of DPDM in the radial direction plays a pivotal role in our calculations due to the spherical symmetry assumption. In our response to **1.1b**), we will address the impact of small-scale density fluctuations and demonstrate that they do not significantly affect our calculations.

1.1b). *Moreover, the conversion probability changes as one considers non-radial trajectories. The latter can potentially also be quite sensitive to small scale inhomogeneities in the plasma, should they exist (in other words, this raises the question: can the conversion surface truly be treated as spherical?). I think a more careful treatment is required.*

Reply. As explained in the reply for **1.1a**), non-radial trajectories are included in our calculation under the assumption of spherically symmetric and hydrostatic solar model. We emphasize that the quiet Sun model we employ yields satisfactory agreement with various observations [3]. In the following, we estimate the influence arising from small-scale inhomogeneities in the plasma by incorporating density fluctuations.

In our calculations, density fluctuation generally can induce three effects: (1) it can modify the magnitude of the $A' \rightarrow \gamma$ conversion probability by

altering $|\vec{\nabla}\omega_p|$; (2) it can introduce non-spherical modifications to the conversion surface, as suggested by the referee; and (3) the plasma's inhomogeneity induces scattering and absorption of the converted photons, resulting in smearing of their velocity directions and a reduction in photon flux. We have addressed the third effect in the Propagation Section of the manuscript, utilizing the Monte Carlo ray-tracing method.

As we will see below, for Point (1), the inclusion of density fluctuations will induce two new effects which behave oppositely in modifying the conversion probability, $P_{A'\rightarrow\gamma}$. One is that the derivative of electron density with respect to distance becomes larger; the other is that there are more resonant points where $\omega_p(r) = m_{A'}$. It turns out that the two effects cancel out each other, and thus $P_{A'\rightarrow\gamma}$ with the inclusion of density fluctuations remains the same as the original one. In addition, the non-spherical effect in Point (2) is insignificant. Next, we will quantify the effects of (1) and (2).

In Ref. [9], the authors employed an advanced Monte Carlo simulation technique to address density fluctuations and their impact on photon refraction and scattering during plasma propagation. Their findings suggest that refraction and scattering may be the primary causes for the observed lower brightness temperature in quiet-Sun radio emissions across different frequencies, deviating from expected values. Hence, we closely adhere to their mathematical description of density fluctuations and incorporate it into our own research.

First and foremost, we emphasize that the density fluctuation in the plasma density is relatively small, with an approximate magnitude of [9]

$$\epsilon_e \equiv \Delta n_e/n_e \approx 10\%, \quad (4)$$

and importantly, this fluctuation fraction remains constant as the radial distance changes [9, 10].

The probability distribution of plasma density fluctuations is described by the spatial power spectrum. For the solar corona of the quiet Sun, the spatial power spectrum of density fluctuations can be expressed as [11, 12]

$$P(q) = C_N^2 q^{-\alpha}, \quad q_o < q < q_i, \quad (5)$$

where C_N^2 is the structural constant, q represents the spatial wavenumber, and α corresponds to the power-law exponent, which is chosen as $\alpha = 11/3$ to reflect the Kolmogorov spectrum.

The scale of density turbulence is defined as $l \equiv 2\pi/q$, with $l_i = 2\pi/q_i$ and $l_o = 2\pi/q_o$ denoting the inner and outer scales of the density turbulence, respectively. It is reasonable to assume that $l_o \approx 10^6 l_i$ [9]. Consequently, the steep shape of the Kolmogorov-type spectrum for $P(q)$ indicates that density fluctuations predominantly occur on larger scales.

The inner scale, denoted as l_i , can be associated with the ion inertial scale, given by the expression

$$l_i = \frac{684}{\sqrt{n_e/\text{cm}^{-3}}} \text{ km.} \quad (6)$$

Consequently, for the plasma layers corresponding to frequencies of 30, 40, 60, and 80 MHz, the respective inner scales l_i are estimated to be 0.2 km, 0.15 km, 0.1 km, and 0.075 km.

The spatial power spectrum can be normalized to the variance of the density fluctuations $\langle \Delta n_e^2 \rangle \equiv (\epsilon_e n_e)^2$ as,

$$\int_{q_o}^{q_i} P(q) 4\pi q^2 dq = \langle \Delta n_e^2 \rangle = (\epsilon_e n_e)^2. \quad (7)$$

Using the Kolmogorov spectrum with $\alpha = 11/3$, it can be determined that

$$C_N^2 = \frac{q_o^{\alpha-3}}{6\pi} \langle \Delta n_e^2 \rangle, \quad (8)$$

$$P(q) = \frac{q_o^{\alpha-3}}{6\pi} q^{-\alpha} \langle \Delta n_e^2 \rangle. \quad (9)$$

The fluctuations can be expressed in the Fourier modes,

$$\Delta n_e(r) = \int_1^{\tilde{q}_i} d\tilde{q} \Delta \tilde{n}_e(q) e^{iqr}. \quad (10)$$

We have rescaled the momentum $\tilde{q} \equiv q/q_o$ for convenience. Subsequently, the fluctuations averaged over the length scale $l_o = 2\pi/q_o$ are

$$\begin{aligned} \langle \Delta n_e^2 \rangle &= \frac{1}{l_o} \int dr \int_1^{\tilde{q}_i} d\tilde{q} \int_1^{\tilde{q}_i} d\tilde{q}' \Delta \tilde{n}_e(q) \Delta \tilde{n}_e(q') e^{i(q-q')r} \\ &= \int_1^{\tilde{q}_i} d\tilde{q} \langle \Delta \tilde{n}_e^2(q) \rangle. \end{aligned} \quad (11)$$

Compared with Eq. (7), we have the fluctuations in the momentum space,

$$\langle \Delta \tilde{n}_e^2(q) \rangle = P(q) 4\pi q^2 q_o = \frac{2}{3} \tilde{q}^{2-\alpha} \langle \Delta n_e^2 \rangle. \quad (12)$$

The averaged derivative of $n_e(r)$ with respect to r , in the squared form, is then

$$\begin{aligned}\langle (n'_e)^2 \rangle &\simeq \langle (\Delta n'_e)^2 \rangle = \frac{1}{l_o} \int dr \int_1^{\tilde{q}_i} d\tilde{q} \int_1^{\tilde{q}_i} d\tilde{q}' \Delta \tilde{n}_e(q) \Delta \tilde{n}_e(q') q q' e^{i(q-q')r} \\ &= \int_1^{\tilde{q}_i} d\tilde{q} \Delta \tilde{n}_e^2(q) \cdot q^2 \simeq \frac{2}{3} \langle \Delta n_e^2 \rangle \frac{1}{5-\alpha} q_o^2 \tilde{q}_i^{5-\alpha}.\end{aligned}\quad (13)$$

In the first step, we have omitted the derivative of the background electron density, $n_{e,\text{bkg}}(r)$, which is small compared with that of the fluctuations. The averaged second derivative of $n_e(r)$ with respect to r , in the squared form, is then

$$\begin{aligned}\langle (n''_e)^2 \rangle &\simeq \langle (\Delta n''_e)^2 \rangle = \frac{1}{l_o} \int dr \int_1^{\tilde{q}_i} d\tilde{q} \int_1^{\tilde{q}_i} d\tilde{q}' \Delta \tilde{n}_e(q) \Delta \tilde{n}_e(q') q^2 q'^2 e^{i(q-q')r} \\ &= \int_1^{\tilde{q}_i} d\tilde{q} \Delta \tilde{n}_e^2(q) \cdot q^4 \simeq \frac{2}{3} \langle \Delta n_e^2 \rangle \frac{1}{7-\alpha} q_o^4 \tilde{q}_i^{7-\alpha}.\end{aligned}\quad (14)$$

Next, we are going to check whether the WKB approximation and saddle-point approximation are threatened by the above density fluctuations.

Using Eq. (13), we can estimate the typical length scale of density variations as

$$\delta l_e = \left| \frac{n'_e}{n_e} \right|^{-1} \simeq \left[\sqrt{\frac{2}{3}} \left(\frac{1}{5-\alpha} \right)^{\frac{1}{2}} \epsilon_e q_o \tilde{q}_i^{\frac{5-\alpha}{2}} \right]^{-1} \simeq 10^{-3} q_o^{-1}, \quad (15)$$

which turns out to be much larger than the dark photon wavelength, $\delta l_e k_{A'} \approx 30 \gg 1$. Therefore, the WKB approximation we applied in deriving Eq. (3) is justified even with the density fluctuations included.

Another important length scale is the resonant conversion length, $\delta l_{\text{res}} = \sqrt{2\pi/F''(r_c)}$. It is defined as the length along which the phase factor (see Eq. (17) in the new manuscript), $F(r) \equiv \int dr [\omega_p^2(r) - m_{A'}^2]/2k_{A'}$, changes by π . This is the length interval which dominantly contributes to $P_{\gamma \rightarrow A'}$. We have

$$\delta l_{\text{res}} = \sqrt{2\pi \frac{k_{A'}}{\omega_p} \frac{1}{\omega'_p}} \simeq \sqrt{2\pi} v_{\text{DM}} \left(\frac{\delta l_e}{k_{A'}} \right)^{1/2}, \quad (16)$$

which obviously satisfies $\delta l_{\text{res}} \ll \delta l_e$.

Next, we evaluate the robustness of the saddle-point approximation. The crucial criterion is that the second derivative $F''(r)$ plays a dominant role in

the Taylor series of $F(r)$ compared with the higher derivative terms (note that at the resonant point, $F'(r) = 0$). Then we calculate the following quantity with the help of Eqs. (13) and (14),

$$\frac{\frac{1}{2!}F''(r)}{\frac{1}{3!}\delta l_{\text{res}}F'''(r)} \simeq \frac{3}{\sqrt{2\pi}} \frac{[F''(r)]^{3/2}}{F'''(r)} \simeq \frac{3}{\sqrt{2\pi}} \left(\frac{1}{2k_{A'} n_e} \frac{\omega_p^2}{n''(r)} \right)^{1/2} \frac{[n'(r)]^{3/2}}{n''(r)} \simeq 5. \quad (17)$$

We see that the second derivative indeed plays a dominant role, and thus we consider the saddle-point approximation still holds true with an acceptable accuracy. Also, we notice that Ref. [13] numerically shows that the saddle-point approximation works well when the ratio in Eq. (17) is larger than unity.

The above arguments show that the form of conversion probability, Eq. (3), is still correct with the density fluctuations considered. However, its numerical value may be altered by the inclusion of density fluctuations. But as we are going to demonstrate below, the value of $P_{\gamma \rightarrow A'}$ remains the same as the original one. Now, we have two new effects that behave oppositely in modifying the value of the probability $P_{A' \rightarrow \gamma}$. On one hand, the derivative of n_e with respect to distance at the resonant points becomes larger upon the inclusion of density fluctuations, which lowers $P_{A' \rightarrow \gamma}$. On the other hand, we have more resonant points (> 1) where $\omega_p(r) = m_{A'}$ due to the density fluctuations, which increases $P_{A' \rightarrow \gamma}$. We show an example of $n_e(r)$ profile in Fig. 1 where the effects of a larger derivative and more intersections can be seen. We use r_{dn} to denote the ratio between $P_{A' \rightarrow \gamma}$ with and without density fluctuations, and it can be calculated as

$$r_{\text{dn}} = \frac{\sum_{n_e(r')=n_e(r_c)} \left| \frac{1}{n_e(r)} \frac{dn_e(r)}{dr} \right|_{r=r'}^{-1}}{\left| \frac{1}{n_{e,\text{bkg}}(r)} \frac{dn_{e,\text{bkg}}(r)}{dr} \right|_{r=r_c}^{-1}} = \frac{\sum_{n_e(r')=n_e(r_c)} \left| \frac{dr}{dn_e(r)} \right|_{r=r'}}{\left| \frac{dr}{dn_{e,\text{bkg}}(r)} \right|_{r=r_c}}. \quad (18)$$

We can first make a rough estimate of r_{dn} . Suppose the fluctuation amplitude is δn_e in a length scale δr . The intersections can only occur within the interval Δr along which the background density $n_{e,\text{bkg}}(r)$ changes by δn_e . The number of intersections can be estimated as $\Delta r/\delta r$. Then, we have $r_{\text{dn}} \sim \frac{(\Delta r/\delta r) \cdot (\delta r/\delta n_e)}{\Delta r/\delta n_e} \sim 1$. We see that the two effects cancel out each other. Thus, we anticipate that the average conversion probability, when we include the density fluctuations, will remain the same as our original value.

We then numerically compute the ratio r_{dn} in Eq. (18) with a large sample of n_e profile generated by Monte Carlo method. As we will see below, the concise result $r_{\text{dn}} \simeq 1$ is indeed verified.

For the convenience of numerical computation, we first need to discretize the density fluctuations, Eq. (7), as

$$\begin{aligned}
\langle \Delta n_e^2 \rangle &= \frac{2}{3} \langle \Delta n_e^2 \rangle \int_0^{\log_{10} \tilde{q}_i} \tilde{q}^{3-\alpha} \ln(10) \cdot d \log_{10} \tilde{q} \\
&= \frac{2}{3} \langle \Delta n_e^2 \rangle \sum_{n=0}^N \tilde{q}^{3-\alpha} \cdot \Delta \cdot \ln(10) \\
&= \frac{2 \ln(10)}{3} \sum_{n=0}^N \langle \Delta n_e^2 \rangle^{(\text{disc.})}(q_n) \cdot \Delta
\end{aligned} \tag{19}$$

where

$$\langle \Delta n_e^2 \rangle^{(\text{disc.})}(q_n) \equiv \langle \Delta n_e^2 \rangle \tilde{q}_n^{3-\alpha}, \quad q_n = 10^{n\Delta} q_o, \quad \tilde{q}_n = 10^{n\Delta}. \tag{20}$$

$\langle \Delta n_e^2 \rangle^{(\text{disc.})}(q_n)$ is the variance (squared) of density fluctuations of the q_n mode in the interval $[\log_{10}(q_n/q_o), \log_{10}(q_n/q_o) + \Delta]$. The discretization is carried out in the logarithmic scale, as the momentum span is broad, spanning 6 orders of magnitude from q_o to q_i . The total number of modes is $N = \log_{10}(q_i/q_o)/\Delta$. Next, the variance of density fluctuations for different momentum modes can be estimated as

$$\sigma_{n_e}(q_n) \simeq \sqrt{\langle \Delta n_e^2 \rangle^{(\text{disc.})}(q_n)} \simeq \epsilon_e n_e \left(\frac{q_n}{q_o} \right)^{\frac{3-\alpha}{2}}. \tag{21}$$

The density as a function of distance, $n_e(r)$, with the fluctuations taken into consideration, can be modeled as

$$n_e(r) = \frac{r_c^2}{r^2} \left(n_{e,\text{bkg}}(r_c) + \sum_{n=0}^N \delta n_e(q_n) \Delta \cdot \sin[q_n(r - r_c) + \phi_n] \right). \tag{22}$$

Note that we have used sine functions for simplicity in numerical evaluation. We also add random phases ϕ_n for each mode. Based on (21), we have the variance of $\delta n_e(k_n)/n_e(r_c)$,

$$\frac{\sigma_{n_e}(k_n)}{n_e(r_c)} \simeq \epsilon_e \cdot 10^{n\Delta \cdot (\frac{3-\alpha}{2})}. \tag{23}$$

We then use Monte Carlo method to generate the values of $\delta n_e(k_n)$ following a Gaussian distribution with the mean value at zero and the variance $\sigma_{n_e}(k_n)$ for each k mode. Also, the phase ϕ_n randomly picks up a value between 0 and

2π for each k mode. To check the effect of the fluctuations on the conversion probability of dark photon to photon, we take the frequency 40 MHz as an example. At this frequency, the solar wind dominates so that the background density profile can be taken as r^{-2} as shown in Eq. (22). In this example, the corresponding resonant conversion layer is at $r_c \simeq 9.4 \times 10^5$ km from the solar center, and therefore $q_0 r_c \simeq 40$.

Figure 1: The density profile exhibiting fluctuations depicted by the blue line. This is plotted with the first 12 modes included. This profile is centered around the resonant layer corresponding to 40 MHz. The electron density $n_e(r_c)$ for the 40 MHz frequency is illustrated by the orange line.

To proceed, we take $\Delta = 0.2$ and thus we have $N = 30$ modes in total. Computing r_{dn} in the presence of the full $N = 30$ modes turns out to be challenging due to numerical limitations. Consequently, we take the first 2, 4, 6, ..., 20 modes respectively. For each number of modes we choose, we iteratively evaluate r_{dn} 1000 times with different Monte Carlo n_e profiles and then take the average to ensure statistical stability. In Fig. 2, we present the result of r_{dn} with more modes gradually included in computations. We see that the average value of r_{dn} converges to approximately 1, insensitive to the number of modes. Therefore, we conclude that including density fluctuations does not significantly change the value of $P_{A' \rightarrow \gamma}$, as the two effects of larger derivatives and more resonant points cancel out each other.

Then, we check Point (2) which is about the modification to the shape of the conversion surface. The deformation of the conversion surface is within the length interval Δr around r_c . As we can see from Fig. 1, $\Delta r/r_c \ll 1$.

Figure 2: The ratio r_{dn} , representing the relationship between the modified variation of n_e and its original value of 0.5, is computed numerically for various numbers of modes k at the 40 MHz frequency, and is shown as the blue dots, while the orange line denotes the original value with $r_{\text{dn}} = 1$. These calculations are performed over a total of 1000 samples and the resulting values are averaged.

Therefore, the deformation effect is negligible compared with the original conversion sphere located at $r = r_c$ without density fluctuations included.

To summarize, we have provided a quantitative demonstration that inhomogeneities have a minimal impact on our calculations. This conclusion holds true for the condition of deriving the conversion probability, the magnitude of the conversion probability, and the deformation of the conversion sphere. Two key factors contribute to the result: Firstly, the fraction of density fluctuation remains small, at approximately 10%. Secondly, the density fluctuation predominantly occurs at larger scales, indicating that small-scale turbulence has a limited effect.

We have summarized the main points above in the Methods section of the new manuscript. Additionally, we plan to further organize this content and include it in a new, separate paper.

1.2). *The authors do not address what the width of the line should. Presumably this means they are neglecting broadening effects? It is not clear to me whether this is valid. In general, this will depend on the velocity of*

the plasma, the scale and shapes of the inhomogeneities, and the number of scatterings (not to mention the broadening induced as the photon accelerates away from the production point). The authors could incorporate this direction into their ray tracing procedure – should they do this, the authors should also be careful to show how the assumptions currently adopted on the scattering surface alter this result. I believe the authors need to more firmly establish that the line will end up in a single bin.

Reply. The signal of converted photons is not perfectly monochromatic but has a finite width, mainly due to the following two effects: one is the velocity dispersion of dark matter itself; the other one is the scatterings of converted photons by inhomogeneities during propagation. The line width induced by dark matter velocity dispersion is [1]

$$B_{\text{sig}} \approx \frac{m_{A'} v_{\text{DM}}^2}{2\pi} \sim 130 \text{ Hz} \frac{m_{A'}}{\mu\text{eV}} \quad (24)$$

where $m_{A'}$ is the dark photon mass and $v_{\text{DM}} \sim 10^{-3}c$ is the typical velocity of dark matter. For the frequency range $m_{A'} \sim 10^{-7}\text{eV}$ we are interested in, such a signal width (e.g. 13 Hz) is much smaller than the LOFAR frequency resolution (i.e., binwidth) which is 97 kHz. Therefore, if only considering the effect of dark matter velocity dispersion, the signal line is safely in a single bin. Next, we discuss the second effect, that is, the scatterings by inhomogeneities. We find that the line width due to this effect is still far smaller than the LOFAR binwidth.

After being converted from dark photons, the photons propagating in the solar plasma will experience scatterings by plasma inhomogeneities, which can be described by a Fokker-Planck equation [14, 15]. Such scatterings are essentially the refraction of photons due to the fluctuations of refractive index determined by the plasma density [14, 15]. The scatterings will result in angle dispersion and energy dispersion of photons. The effect of angle dispersion has already been included in the ray-tracing code [14], which is described by the angular distribution function $g(\theta_1, \phi_1)$ and further $\beta(f)$ in our paper. When considering the angle distribution, the scatterings by density fluctuations are assumed to be elastic so there is no energy dispersion. This is reasonable because the density fluctuations are effectively static with a velocity far smaller than the speed of light [14].

However, the velocity of density fluctuations may become important for photons with the smallest velocities just after conversion, when the photon effective mass is equal to the local plasma frequency. Therefore, we must be

careful to check the velocity of density fluctuations. The typical density fluctuation is the ion-sound waves [15] with the speed $C_s \sim \sqrt{[T_e(1 + 3T_i/T_e)]/m_i}$ (see e.g., Ref. [16]) where T_e is the electron temperature; T_i and m_i are respectively the ion temperature and ion mass. For the typical value $T_e \sim 10^6$ K at the conversion layer, we have $C_s \sim 100$ km/s. Therefore, the photon velocity modification due to scatterings near the conversion layer is estimated to be around 100 km/s, which is comparable with the dark matter velocity $v_{\text{DM}} \sim 10^{-3}c$ in (24), and cannot broaden the line width significantly. As a result, the line is still safely in a single LOFAR frequency bin.

In the context of plasma scattering from inhomogeneities, photons emitted away from the production point experience much lesser frequency alterations compared to those emitted at the production point. This is because the electron density rapidly decreases as the photons propagate out. Also, plasma velocity quickly becomes negligible compared to the velocity of photons which quickly become relativistic as they leave the resonant layer. Nonetheless, the scattering process can still significantly modify photon directions due to the nature of refraction [14, 15]. In our study, we account for this effect in the Propagation section using the Monte Carlo ray-tracing technique, as outlined in Ref. [14]. Additionally, these arguments elucidate why Ref. [14] only considers photon direction smearing caused by scattering in the inhomogeneous plasma while neglecting energy smearing.

We have summarized the main points above below Eq. (6) in the new manuscript.

1.3a). *I have a number of concerns regarding the authors statistical treatment. First, why have the authors chosen to use $k=10$ and $n=3$? What is the impact of changing these to other reasonable values?*

Reply. The upper limits on mono-chromatic signals, obtained from the log-likelihood ratio test, exhibit robustness against variations in background fitting parameters. These parameters, denoted as n for the degree of the polynomial function and k for the number of bins included in the calculation, do not significantly affect the results. Typically, one employs quadratic and trilinear forms for the polynomial function, and even with these different choices, the outcome remains largely unchanged. To demonstrate this resilience, we conducted a comprehensive analysis on LOFAR data recorded on September 3, 2015, using varying degrees of polynomials and numbers of adjacent bins. Specifically, we examined three cases: 10 adjacent bins with a 3rd-degree polynomial, 10 adjacent bins with a 2nd-degree polynomial, and 8 adjacent bins with a 3rd-degree polynomial. The results of these analyses

Figure 3: The 95% C.L. upper limits from LOFAR data on September 3, 2015 with a constant mono-chromatic signal using different background fitting parameters. The orange, cyan and blue limits represent using 10 adjacent bins with 3rd-degree polynomial, 10 adjacent bins with 2nd-degree polynomial and 8 adjacent bins with 3rd-degree polynomial, respectively.

are presented in Fig. 3.

Our investigation demonstrates that the derived signal limits remain remarkably stable and impervious to the choices of different n and k . This robustness serves to reinforce the reliability and consistency of our method in establishing upper limits on the mono-chromatic signal from the LOFAR data.

We have summarized the main points above in the Methods section of the new manuscript.

1.3b). *Can the authors show samples of how well these fits work, and what the residuals look like? Are the residuals Gaussian? If not, then it doesn't make sense to adopt the Gaussian approximation.*

Reply. In our fitting process, the flux $F(t_i, f_j)$ is characterized by its time index t_i and frequency index f_j . To analyze each frequency bin f_j , we calculate the average flux over time, denoted as $\bar{F}(f_j)$, and assume it varies smoothly in frequency, fitting it using 3rd-degree polynomials. For a fixed frequency bin, we consider the fluxes of different time bins to follow a Gaussian distribution, with $\bar{F}(f_j)$ serving as the mean of the Gaussian function.

Figure 4: The flux distribution of LOFAR data from time-bins after cleaning process on September 3, 2015, is presented in the top-left panel for $f_{200} \approx 49.21$ MHz and bottom-left panel for $f_{400} \approx 68.74$ MHz, which are well-fitted to Gaussian distribution. Additionally, on the right panel, we display the 3rd-degree polynomial fittings to the time-averaged flux in the time-bins, \bar{F} , in the adjacent frequency bins for data at f_{200} in the top-right panel and f_{400} in the bottom-right panel.

To validate the assumption of Gaussian distribution, we specifically examine two frequency bins ($j = 200, 400$), corresponding to approximately 49.21 MHz and 68.74 MHz on September 3, 2015, respectively. After undergoing the cleaning process, each bin yields 920 and 1040 time bins remaining. The top-left and bottom-left panels of Fig. 4 display the histograms for the flux at f_{200} and f_{400} , respectively, and these plots align well with the Gaussian distribution. The residuals, obtained by subtracting the mean value, will exhibit the Gaussian distribution as expected.

Furthermore, in the right panels of Fig. 4, we present the time-averaged flux $\bar{F}(f_j)$ of the 11 adjacent frequency bins and fit to the data with a 3rd degree polynomial function as an example, demonstrating an excellent fit to the background flux.

Furthermore, to be conservative, when setting the upper limits we include the deviations of the time-averaged flux $\bar{F}(f_j)$ from the background fitting

line in the right panels as systematic errors, as we have discussed in the paragraph above Eq. (8) in the new manuscript.

We also show the above histograms in the Methods section of the new manuscript.

1.3c). *Its also unclear to me whether it is really fair to adopt the half-chi2 distribution – this is not always a valid assumption, especially if one is dominated by large non-Gaussian systematics. The authors should provide many more details and checks to show that their procedure does in fact reflect the stated sensitivity.*

Reply. Following the methodology presented in Ref. [17, 18], we employed a Monte-Carlo (MC) procedure to assess the suitability of Wilk’s theorem for our data. Specifically, for a particular frequency bin, we constructed MC data under the null hypothesis using the expression $\mathbf{d}_{\text{MC}} = \{B(f_i) + S\delta_{i i_0}, \sigma_{\text{tot}}^i\}$, where $B(f)$ represents the fitted background function, and σ_{tot} corresponds to the total uncertainty derived in our paper. Each set of MC fake data comprises 11 bins centered around frequency f_{i_0} , encompassing 10 adjacent bins.

To evaluate the test statistics q_S for each group of MC data, we generated 10,000 such groups. Focusing on the 200th frequency bin as an illustrative example, we plotted the distribution of test statistics q_S in Fig. 5. Additionally, we presented the corresponding half-chi-squared distribution for comparison.

Through this comparative analysis, we observed that the test statistics q_S follows a half- χ^2 distribution, indicating that Wilk’s theorem is well-suited for our data.

1.4a). *Finally, I have two smaller comments: 1.) I find the results on the axion rather uninteresting – this is not even remotely competitive, and I suppose that the authors knew this before preceding in this direction. I personally find that including axions tends to distract from the most important message, and I would consider putting it in an appendix, or at the very least de-emphasising the result (leaving the focus on the dark photon).*

Reply. In response to the referee’s comment, we have made the necessary adjustments to the presentation of our results. Specifically, we have relocated the results pertaining to the axion case (Fig.10) from the main text to the “appendix,” specifically under the section titled *Constraint on axion-like particle dark matter* in the Methods section. Furthermore, we have also transferred the introduction, formulas, and discussions related to the axion

Figure 5: The distribution of test statistics q_S for MC generated data. The solid blue line shows the one-degree half-chi-squared distribution, while the red histogram line represents the distribution of q_S obtained from 10,000 MC fake data sets.

case from the main text to the same section, along with Fig.10.

Furthermore, in accordance with the suggestion made in **2.5)** of **Referee 2**'s comments, we have re-plotted Fig.10, this time using a smaller magnetic field, which is now more appropriately represented as a function of distance from the solar center.

1.4b). 2.) *At the moment the text does not read remarkably well. Articles are frequently missing and sentences seem to be put together in a bit haphazard manner. I would request that the authors dedicate time into improving this.*

Reply. We apologize for the lack of articles and disorganized sentence structure in the current text and appreciate the referee's feedback. We have devoted time to enhancing the manuscript's clarity and coherence, striving to make the text more readable and ensure a high-quality presentation in the new version.

Replies to Referee 2

2.1). *The use of acronyms that are not defined is rife throughout the paper (e.g. WIMP, CP, QCD, SM, CMB, DPDM). All acronyms should be defined the first time they are used.*

Reply. We appreciate the reviewer's valuable feedback, which has led to improvements in the manuscript. We have thoroughly addressed the issue by revising our paper to ensure that all acronyms, such as WIMP, CP, QCD, SM, CMB, and DPDM, are now appropriately defined the first time they appear in the text. We hope the modifications have significantly enhanced the clarity and comprehensibility of our manuscript.

2.2). *There are also a number of symbols used in the equations and the text that are not defined (e.g. m_A , A , μ , m_{DM} ...). All symbols should be defined the first time they are used.*

Reply. We thank the referee for bringing this concern to our attention. We have made necessary revision to our paper. Now, all symbols used in the equations and the text, including m_A , A , μ , m_{DM} , have been defined the first time they are used.

2.3). *The formula for plasma frequency is incorrectly in units of eV. There is an extra $1/(hc)$ present in the formula for angular plasma frequency.*

Reply. In our paper, we adopt the conventional practice of using natural units, where fundamental constants such as \hbar and c are set to 1 in particle physics calculations. In this notation, the relations between the energy E , angular frequency ω and frequency f are

$$E = \hbar\omega = hf, \quad \text{with } \hbar \rightarrow 1, \quad h \rightarrow 2\pi. \quad (25)$$

The plasma frequency formula presented in the manuscript is consistent with the usage of natural units.

To ensure clarity and avoid any misunderstandings, we explicitly stated below Eq. (2) in the new manuscript that we are employing natural units. This information was declared in advance to avoid misleading for the readers.

2.4). *The model that has been used for the quiet Sun is based upon millimeter and submillimeter wavelengths (GHz frequencies), whereas the wavelengths*

used for the LOFAR analysis are between 1-10 m. The authors should use a standard solar density/frequency model for the Sun that has been derived from relevant wavelengths to the LOFAR observations.

Reply. We thank the referee’s advice regarding the solar profile for the quiet Sun. In our research, we focused on the quiet sun and adopted a spherical symmetric and hydrostatic model, where the gas pressure is in equilibrium with gravitational force, remaining static over time.

To construct our model, we utilized the work by Ref. [4], which derived temperature (T) and hydrogen density (n_H) profiles for the quiet Sun. It is based on the photospheric model from Ref. [2] and the coronal model from Refs. [5, 6]. The profiles were calculated by various research groups [7] using a chromosphere model from Ref. [8] and the coronal model from Refs. [5, 6]. Importantly, these independent calculations consistently agree with one another and have been validated by observations of atomic lines in the soft X-ray range [3] and extreme ultraviolet range [2]. Therefore, the hydrostatic equilibrium of the quiet Sun region has been confirmed in previous studies [2, 3] but at higher frequencies, as pointed out by the referee.

Figure 6: The comparison between the solar profile we used and the profile LOFAR observed with a hydrostatic density model in our frequency range. The x-axis represents the radius of the sun.

As a result, following the suggestion from the referee, we conducted a comparison of the solar profile we utilized [4] to the one which fits to the solar intensity profile observed by LOFAR in the frequency range of 30 – 80 MHz using a ray-tracing simulation technique [19]. Ref. [19] has shown that employing the hydrostatic model can provide a good fit to their data in high

frequency range. Although they found that in low frequency range it needs to incorporate supersonic solar wind, the difference is small. We have compared the solar profile in Ref. [4] and the profile from LOFAR fitting in Ref. [19] in our frequency range, and plotted the two in Fig. 6. The deviation between the two profiles is within a factor of two, thus the effect on the final photon signal is not large.

We have added the discussion of the solar model in the Methods section.

2.5). *The value of the magnetic field chosen is too high for the solar corona at the heights that correspond to LOFAR frequencies. These heights are typically 1.5 to 2 solar radii in altitude, whereas the estimates of magnetic field are taken between 1.05-1.35 solar radii. In addition, the magnetic field of the Sun has a strong dependence with distance, dropping off at or faster than the $1/r^3$ dependence one gets in a standard dipole. The authors should consider this magnetic field dependence, given the importance of B in the calculation of axion conversion.*

Reply. We thank the referee for pointing out this imprecision. For our frequency range, 30-80 MHz, the resonant conversion happens at the distance 1.51-1.13 R_\odot from the solar center. This is obtained based on the relation between plasma frequency and electron number density (please refer to Eq.(2) in the new manuscript) and the profile of solar electron number density as a function of altitude (please refer to Fig.4 in the new manuscript). We then conservatively re-estimate the value of the magnetic field with the following paragraph added in the new manuscript (in *Constraint on axion-like particle dark matter* in the Methods section):

“...In our case, the resonant conversion happens at about the range of 1.51-1.13 R_\odot (corresponding to frequencies in the range of 30-80 MHz). To proceed conservatively, we estimate $|\mathbf{B}_T|$ to be 1 Gauss at $1.05R_\odot$ and extrapolate this value to obtain $|\mathbf{B}_T| \approx 0.34\text{-}0.79$ Gauss for our frequency range, following the attenuation relation $\propto R^{-3}$.”

Then, we re-plot the constraint on axion (see Fig.10 in the new manuscript) with $|\mathbf{B}_T|$ set as a function of distance based on the extrapolation above. Due to the large inhomogeneities of the coronal magnetic field which depend on location and time, it is difficult to calculate the axion-photon conversion rate exactly. Consequently, we make the above estimate to set a conservative upper limit for the axion case. Additionally, as suggested by **1.4a**), we have moved Fig.10 from the main text to the “appendix” (i.e., *Constraint on axion-like particle dark matter* in the Methods section). This is to prevent

readers from becoming distracted by the axion case, which does not provide any constraints beyond the most stringent existing ones and is therefore less intriguing than the dark photon case. Correspondingly, we have also moved the brief introduction, formulas, and discussions related to the axion case from the main text to the Methods section, together with Fig.10.

2.6). *The most probable velocity of 235 km/s at the conversion layer seems oddly specific. Some more information should be given about how/why this velocity is chosen.*

Reply. Based on the Standard Halo Model for dark matter, the velocity of dark matter particles follows a Maxwell distribution with the highest probability, denoted as v_p , taken as the rotational speed of the local standard of rest around the Galactic center. Once boosted to the rest frame of the Sun, v_p can be considered the typical velocity of dark matter relative to the Sun (see e.g., Refs. [20]). Different models predict slightly different values for v_p , typically falling within the range of 210 – 270km/s (see e.g., Ref. [21, 22]). The specific value given by the model in Ref. [21] is $v_p \simeq 235$ km/s, with an uncertainty of ± 19 km/s. However, this uncertainty will not affect the final uncertainty of constraints on ϵ . This is because the velocity cancels out when calculating the power of converted photons, as shown by substituting Eq. (4) into Eq. (5) in the manuscript.

2.7). *When talking about the scattering effect, at line 142, it should be made clear that the authors are discussing the tied-array beam mode that was used for the observations in question. Standard interferometric LOFAR observations do not have multiple beams that resolve the source.*

Reply. We appreciate the referee’s advice regarding the description of LOFAR observation mode, and we modified our description in the new manuscript (in the paragraph above Eq. (6))

“ We are using LOFAR data made in the tied-array beam mode. While this mode offers a nice angular resolution [87], the field of view (FOV) of each LOFAR beam is significantly smaller than the total angular span of the Sun. ”

2.8). *Line 195 – this statement is half-true. The frequency resolution for beam-formed imaging is increased but the spatial resolution is significantly decreased.*

Reply. We are grateful for the referee’s valuable advice. We acknowledge

that our original statement may have been misleading. We agree that while the frequency resolution is indeed increased in the beam-formed mode, it is accompanied by a significant decrease in spatial resolution.

To avoid confusion, we have revised the relevant sentences in our paper (in the first paragraph of the subsection *LOFAR data analysis and setting constraints on the ultralight DM couplings* in the Results section):

“This mode offers significantly increased frequency resolution while reducing spatial resolution. ”

2.9). *Figure 2: it is not clear what the increase in intensity that occurs at 40 MHz comes from. The increase looks like potentially bad channels within the LOFAR data. This conclusion is based upon the triple peak that has occurred whenever there is an increase in the flux, and the choice of $n=3$ (line 227). The authors should look over the data and double check the raw LOFAR data before the sub-bands are created from the individual channels. The intensity does not look like solar radio bursts, as indicated from line 240. These should have been filtered out via the data cleaning, and are typically broader in bandwidth.*

Reply. We sincerely appreciate the referee’s valuable observation and insightful comments regarding Figure 2 and the potential issues related to the increase in intensity at 40 MHz.

After conducting a thorough reexamination of the data, we concur with the referee’s assessment that the observed increase in intensity at 40 MHz is likely attributable to bad channels within the LOFAR data. To address this concern comprehensively, we have made substantial revisions to the relevant section in the paper.

We now explicitly acknowledge the possibility of bad channels being responsible for the observed behavior at 40 MHz, rather than solar bursts. Consequently, we have excluded the frequency bin associated with the bad channel (the 112th bin with $f \sim 40.6$ MHz) from our procedure of setting limits. The paper has been edited accordingly below Eq. (11) as follows:

“In the 112th frequency bin with $f \sim 40.6$ MHz for all three periods of observations, an increasing intensity was observed. However, this bin was identified as a bad channel and subsequently excluded from our analysis.”

2.10). *Figure 4: The authors should discuss the changes when the magnetic*

field changes with distance. This appears to reduce the gradient of the line, or even reverse the sign depending upon how $B(r)$ is assumed.

Reply. Yes. As we have answered in **2.5**), we used the relation $B \propto R^{-3}$ to extrapolate the value of magnetic field at $1.05R_{\odot}$ to the values at 1.51-1.13 R_{\odot} corresponding to 30-80 MHz. Then, we have re-plotted Fig.10 (that is, Fig.4 in the old manuscript). Indeed, the gradient of the constraint line becomes somewhat smaller as expected, since now the magnetic field is more properly taken as a function of distance. But the tendency of the constraint line is not reversed, and we think this is because the distance interval 1.51-1.13 R_{\odot} is small.

Replies to Referee 3

3.1a). *My main concern is about the treatment of the plasma frequency (therefore the electron density) in the plasma corona. The authors assume a “static and spherical” solar model, which seems a too rough approximation to me. First of all, although I understand the adopted profile was reported in previous publications (in PRL), it would be useful for the reader to have it in the Nature publication too. I would then ask the authors to write a small Appendix with the simplified solar model they use, maybe also providing an analytical expression for the electron density, which can be used to reproduce the gross features of their results.*

Reply. In our study, we focused on the quiet Sun as it exhibits fewer active events such as turbulence and flares. We adopted a spherical symmetric and hydrostatic model for the quiet Sun, where the gas pressure is in equilibrium with gravitational force, remaining static over time. The hydrostatic equilibrium of the quiet Sun region has been confirmed in previous studies, which agrees with various observational data [2, 3].

To construct our model, we utilized the work by Ref. [4], which derived temperature (T) and hydrogen density (n_H) profiles for the quiet Sun. It is based on the photospheric model from Ref. [2] and the coronal model from Refs. [5, 6]. The profiles were calculated by various research groups [7] using a chromosphere model from Ref. [8] and the coronal model from Refs. [5, 6], and these independent calculations consistently agree with one another.

Following the suggestion of the referee, we provide a simple analytical expression parameterized the hydrostatic density model, which can be ex-

pressed in an exponential form. In this simple form, the electron density is modeled by [19]

$$n_e = N_0 \exp(1/(H_0 r)), \quad (26)$$

with parameter

$$H_0 = \frac{k_B T}{0.6 m_p g_\odot R_\odot^2}, \quad (27)$$

where R_\odot is the solar radius, $g_\odot = 274 \text{ ms}^{-2}$ is the gravitational acceleration at the coronal base, and 0.6 times the proton mass gives the average particle mass in the corona [23]. The temperature T is a scale height temperature determined by both electron and ion temperatures. There are two parameters to be determined: the density N_0 and the temperature T . To carry out our calculation, $N_0 = 1.6 \times 10^{11} \text{ m}^{-3}$ and $T = 2 \times 10^6 \text{ K}$ are used. We present a comparison between the profile we used and the profile modeled by Eq. (26) in Fig. 7.

We have added the above discussion in an “appendix” (i.e., the section *The solar model* in the Methods section).

Figure 7: The comparison between the solar profile we used and the profile modeled by simple analytical expression for hydrostatic model. The x-axis represents the radius of the sun.

3.1b). *Second of all, and most important, I would expect the plasma frequency to have strong temporal variations. In fact, the authors for example recognize that the magnetic field “shows strong magnetic field in-homogeneity depending pending on location, altitude, and time”. If the magnetic field*

varies a lot, I would expect also the electron density to vary consistently. If this is not the case, the authors should justify it with data. In any case, I would ask the authors to look for data about the electron density in the solar corona during the 17 minutes of LOFAR data taking.

Reply. Firstly, as discussed in the response to question **1.1b**), the variance in plasma density is relatively small, with $\Delta n_e/n_e \approx 10\%$ [9]. Additionally, the spatial power spectrum of plasma density fluctuations follows the Kolmogorov spectrum [9], represented as:

$$P(q) = C_N^2 q^{-11/3}, \quad q_o < q < q_i, \quad (28)$$

where q represents the spatial wavenumber. Consequently, the density fluctuation contributes primarily to large-scale turbulence, which has minimal impact on our calculations since the resonant conversion of dark photons to photons occurs over short distances.

The temporal variation of density can be accounted for by the non-static density fluctuations. As we have answered in **1.2**) of **Referee 1**. Such kind of variations will slightly broaden the signal bandwidth because the fluctuations have a non-zero velocity. However, the resultant signal bandwidth ~ 10 Hz is much smaller than the LOFAR frequency resolution 96 kHz, so we can still treat our signal monochromatic. Therefore, such temporal variations do not affect our calculations. As for the fluctuations of magnetic field, we are going to answer in detail in **3.6**) below. We summarize the main points as follows. The correlation length of the smallest magnetic inhomogeneities can be estimated as the size of granules, whose smallest size is $l_M \gtrsim 0.1$ arcsec (70 km) [24]. In comparison, the length of resonant conversion is $\delta l_{\text{res}} \sim 3$ km, which is much smaller l_M . Furthermore, one granule typically lasts ~ 10 minutes before dissipating [25], which is much longer than the time needed for dark matter to cross the resonant length with speed $10^{-3}c$. So both the spatial and temporal variations of magnetic fields will not threaten our calculations.

Moving forward, as suggested by the referees (see reply to question **2.4**), we sought electron density data from LOFAR observations. The electron density profile we utilize and the fitted profile from LOFAR data [19] exhibit good agreement, differing by at most 40%. Moreover, the LOFAR data falls within the frequency range of 30 – 80 MHz, which aligns with the range of our constraints. Although the LOFAR group did not provide density fits specifically for the 17-minute data we used, we contend that their fitted profile serves as a reliable time-averaged estimate for the electron density.

3.2). *My second concern is connected to the first one, and it is related to the “poor” treatment of uncertainties for the plasma profile in general. Apart from what I wrote above, I also felt that the authors give the idea that the electron density profile in the solar corona is very well known. This should be better justified if so. For example, I would find useful a small Appendix in which the authors explain how the electron density is measured (for example using Fe XIII lines) and how precisely.*

Reply. As stated in the response to question **3.1a**), we have used the spherical and hydrostatic model which has been confirmed in previous studies [2, 3]. To conduct our calculations, we considered the electron number density (n_e) and temperature (T) profiles from Ref. [4], which derived the temperature (T) and hydrogen density (n_H) profiles for the quiet Sun based on the photospheric model from Ref. [2] and the coronal model from Refs. [5, 6]. The profiles were calculated by various research groups [7] using a chromosphere model from Ref. [8] and the coronal model from Refs. [5, 6]. These independent calculations consistently agree with each other and have been validated by observations of atomic lines in the soft X-ray range [3] and extreme ultraviolet range (EUV) [2]. Specifically, as prominent EUV spectral lines, Fe XIII lines include sets of line ratios which are density-sensitive and are suitable for solar profile measurement [26]. Many studies have used these lines for density diagnostics based on different observations of the quiet Sun, and they are also in good agreement of hydrostatic model [26, 27, 28, 29]. Thus, it demonstrates the simplicity and reliability of the spherical symmetric and hydrostatic solar model employed in this paper.

Moreover, in the research that employed ray-tracing simulations to fit the solar intensity profile observed by LOFAR in the frequency range of 30 – 80 MHz [19]. They have found that employing the hydrostatic model with the form of eq. (26) provided a good fit to their data, and the solar profile they fitted is in accordance with the profile in Ref. [4].

As a result, the difference of n_e profile for the quiet Sun from the various observations is within a factor of a few. It has impact on plasma frequency, which only shifts the location of resonant region, while the derivative of n_e on radius determines the conversion probability. All these uncertainties are small and the effect on the final photon signal is not large.

We have added the above description in the Methods section.

3.3). *I would like the authors to show to me how the transmission coefficient*

changes when $\delta n_e/n_e$ increases. In particular: how much of the signal gets transmitted if (let's say) $\delta n_e/n_e = 0.2 - 0.5$ instead of 0.1? I would expect a sensitive dependence on this quantity, because waves can get more easily reflected back if they can encounter large up-ward fluctuations when they propagate away from the resonant point.

Reply. Firstly, our choice of $\delta n_e/n_e = 0.1$ is well-founded, supported by *Helios* observations [9, 10], and remains consistent across different radial distances.

Next, we investigate the effects of varying $\delta n_e/n_e$ below. In our analysis of photon propagation in the solar corona, we consider the survival probability P_{sur} , which incorporates inverse bremsstrahlung and Compton scattering, leading to a shift in photon energy. Additionally, the smearing factor β accounts for refraction and scattering, resulting in the bending of photon trajectories. The effect of the transmission coefficient arising from total reflection is encompassed within our smearing factor β . To explore this further, we performed simulations with three different values of $\delta n_e/n_e$ and plotted the corresponding β and P_{sur} , as depicted in Fig. 8.

It is worth noting that both β and P_{sur} decrease as $\delta n_e/n_e$ increases. From the simulation results in Fig. 8, we observed that even for an unrealistic value of $\delta n_e/n_e = 0.5$, the impact on the smearing and survival probability is less than a factor of 2. Furthermore, as we addressed in the response to **3.1**), the small-scale variance in electron density $n_e(r)$ is quite small compared to its large-scale variation. As a result, the scattering and refraction effects from small inhomogeneities are not significant.

Figure 8: The smearing factor β (left) and the survival probability P_{sur} (right) as functions of frequency f , using different values of density fluctuation $\delta n_e/n_e$.

3.4). *The treatment of the background in the likelihood analysis seems a bit arbitrary and no discussion is given to show how sensitive the results are to the choice of a simple polynomial function with $n = 3$. The authors should provide other examples and show how their final plots change accordingly. For example, they can consider a combination of an exponential sine squared kernel (to capture highly oscillatory features in the data) and an exponential-squared kernel (to avoid mismodeling features which do not have strong periodicity). Moreover they should also address the question on how results change varying the considered number of adjacent bins.*

Reply. We have added a discussion about the choice of fitting functions and the number of adjacent bins in the response to **1.3c**. We have tried different functions and numbers of bins, and the results are not sensitive to these choices. Moreover, our signal is narrowly located in a single bin; therefore, the analysis only requires a few adjacent frequency bins. Since there is no prior information about the background, simple polynomial functions are generally suitable choices.

3.5). *The fact that the TS follows a half- χ^2 distribution is an assumption, as far as I can see. It should then be validated through MC simulations.*

Reply. In the response to question **1.3c**, we utilized Monte Carlo simulations to examine the asymptotic behavior of the test statistics, and it was found to approximately follow the half- χ^2 distribution.

3.6). *For the axion case, I have some concern about the conversion probability. Do the authors know if the magnetic field in the solar corona has a strong turbulent component and/or large variations on small length scales? If this is the case, I am not sure the Landau-Zener approximation, usually adopted to derive the conversion probability, holds in this case.*

Reply. The solar coronal magnetic field is highly inhomogeneous. However, as we are going to show below, even for the inhomogeneities at the smallest scales, the correlation length of the inhomogeneities is still much larger than the resonant conversion length of axion DM to photons. Therefore, our saddle-point approximation in calculating the conversion probability still holds well.

The magnetic field extends from the solar photosphere to the solar corona. At the smallest scales, the solar surface (photosphere) is covered by granules. Each granule is associated with a small-scale magnetic field, leading

to an inhomogeneous structure of the magnetic field. Therefore, the correlation length of the smallest magnetic inhomogeneities can be estimated as the size of granules, whose smallest size is $l_M \gtrsim 0.1$ arcsec (70 km) [24]; see also Ref. [30] which gives the typical size of granules as 1500 km. In comparison, the length of resonant conversion is $\delta l_{\text{res}} \sim 3$ km, which is much smaller than the correlation length of magnetic inhomogeneity l_M , i.e., $\delta l_{\text{res}} \ll l_M$. The resonant conversion length, $\delta l_{\text{res}} = \sqrt{2\pi/F''(r_c)}$, is defined as the length along which the phase factor (see Eq. (17) in the new manuscript), $F(r) \equiv \int dr[\omega_p^2(r) - m_{A'}^2]/2k_{A'}$, changes by π . This is the length interval that dominantly contributes to the conversion probability $P_{\gamma \rightarrow A'}$. If we consider the background profile of electron density n_e , that is, n_e drops with the distance following a power-law, we get $\delta l_{\text{res}} \sim 3$ km; if we include the electron density fluctuations, δl_{res} could be even smaller (please refer to Eq. (16) in our answer to **1.1b**) of **Referee 1**).

Furthermore, one granule typically lasts ~ 10 minutes before dissipating [25], which is much longer than the time needed for axion DM to cross the resonant length with speed $10^{-3}c$.

Consequently, it is reasonable to apply the saddle-point approximation to derive the conversion probability. In addition, a proportion of coronal magnetic field can also originate from large-scale structures such as sunspots on the photosphere. However, the size of these large structures are much larger than that of granules, and thus will not threaten the saddle-point approximation either.

Additionally, the other two Referees have comments related to the constraint on the axion case, and we have made changes correspondingly:

As suggested by **2.5**) of **Referee 2**, we have re-plotted Fig.10 with a smaller magnetic field which now is more properly taken as a function of distance from the solar center.

As suggested by **1.4a**) of **Referee 1**, we have moved Fig.10 from the main text to the ‘‘appendix’’ (i.e., the section *Constraint on axion-like particle dark matter* in the Methods section). This is to prevent readers from becoming distracted by the axion case, which does not provide any constraints beyond the most stringent existing ones and is therefore less intriguing than the dark photon case. Correspondingly, we have also moved the brief introduction, formulas, and discussions related to the axion case from the main text to the Methods section, together with Fig.10.

Summary

In addition to the revised manuscript, we also provide a pdf file that highlights the changes we have made so one can track them easily.

We hope that our answers and updates to the text have satisfactorily addressed the comments and questions raised in the report and that our paper can be accepted for publication in Nature Communications.

Sincerely yours,

Haipeng An, Xingyao Chen, Shuailiang Ge, Jia Liu, and Yan Luo

References

- [1] Haipeng An, Fa Peng Huang, Jia Liu, and Wei Xue. Radio-frequency Dark Photon Dark Matter across the Sun. *Phys. Rev. Lett.*, 126(18):181102, 2021.
- [2] J. E. Vernazza, E. H. Avrett, and R. Loeser. Structure of the solar chromosphere. III. Models of the EUV brightness components of the quiet sun. *Astrophysical Journal, Suppl. Ser.*, 45:635–725, April 1981.
- [3] Markus J. Aschwanden and Loren W. Acton. Temperature tomography of the soft x-ray corona: Measurements of electron densities, temperatures, and differential emission measure distributions above the limb. *The Astrophysical Journal*, 550(1):475–492, mar 2001.
- [4] V. De La Luz, A. Lara, E. Mendoza, and M. Shimojo. 3D Simulations of the Quiet Sun Radio Emission at Millimeter and Submillimeter Wavelengths. *Geofisica Internacional*, 47:197–203, July 2008.
- [5] A. H. Gabriel. A Magnetic Model of the Solar Transition Region. *Philosophical Transactions of the Royal Society of London Series A*, 281(1304):339–352, May 1976.
- [6] Peter Foukal. *Solar Astrophysics*. A Wiley-Interscience publication. Wiley, 1990.

- [7] M. Aschwanden. *Physics of the Solar Corona: An Introduction with Problems and Solutions*. Springer Praxis Books. Springer Berlin Heidelberg, 2006.
- [8] J. M. Fontenla, E. H. Avrett, and R. Loeser. Energy Balance in the Solar Transition Region. I. Hydrostatic Thermal Models with Ambipolar Diffusion. *Astrophysical Journal*, 355:700, June 1990.
- [9] G. Thejappa and R. J. MacDowall. Effects of scattering on radio emission from the quiet sun at low frequencies. *The Astrophysical Journal*, 676(2):1338, apr 2008.
- [10] B Bavassano and R Bruno. Density fluctuations and turbulent mach numbers in the inner solar wind. *Journal of Geophysical Research: Space Physics*, 100(A6):9475–9480, 1995.
- [11] W. A. Coles and J. K. Harmon. Propagation Observations of the Solar Wind near the Sun. , 337:1023, February 1989.
- [12] W. A. Coles, W. Liu, J. K. Harmon, and C. L. Martin. The solar wind density spectrum near the sun: Results from voyager radio measurements. *Journal of Geophysical Research: Space Physics*, 96(A2):1745–1755, 1991.
- [13] Nirmalya Brahma, Asher Berlin, and Katelin Schutz. Photon-Dark Photon Conversion with Multiple Level Crossings. 8 2023.
- [14] Eduard P. Kontar, Xingyao Chen, Nicolina Chrysaphi, Natasha L. S. Jeffrey, A. Gordon Emslie, Vratislav Krupar, Milan Maksimovic, Mykola Gordovskyy, and Philippa K. Browning. Anisotropic radio-wave scattering and the interpretation of solar radio emission observations. *The Astrophysical Journal*, 884(2):122, oct 2019.
- [15] N. H. Bian, A. G. Emslie, and E. P. Kontar. A fokker–planck framework for studying the diffusion of radio burst waves in the solar corona. *The Astrophysical Journal*, 873(1):33, mar 2019.
- [16] J. M. Sullivan, M. Lockwood, B. S. Lanchester, E. P. Kontar, N. Ivchenko, H. Dahlgren, and D. K. Whiter. An optical study of multiple NEIAL events driven by low energy electron precipitation. *Annales Geophysicae*, 26:2435, August 2008.
- [17] Glen Cowan, Kyle Cranmer, Eilam Gross, and Ofer Vitells. Asymptotic formulae for likelihood-based tests of new physics. *Eur. Phys. J. C*, 71:1554, 2011. [Erratum: Eur.Phys.J.C 73, 2501 (2013)].

- [18] Malte Buschmann, Christopher Dessert, Joshua W. Foster, Andrew J. Long, and Benjamin R. Safdi. Upper Limit on the QCD Axion Mass from Isolated Neutron Star Cooling. *Phys. Rev. Lett.*, 128(9):091102, 2022.
- [19] Christian Vocks, Gottfried Mann, Frank Breitling, MM Bisi, B Dabrowski, R Fallows, PT Gallagher, A Krankowski, J Magdalenic, C Marqué, et al. Lofar observations of the quiet solar corona. *Astronomy & Astrophysics*, 614:A54, 2018.
- [20] K. Choi, Carsten Rott, and Yoshitaka Itow. Impact of the dark matter velocity distribution on capture rates in the Sun. *JCAP*, 05:049, 2014.
- [21] Paul J. McMillan and James J. Binney. The uncertainty in Galactic parameters. *Mon. Not. Roy. Astron. Soc.*, 402:934, 2010.
- [22] Jo Bovy, David W. Hogg, and Hans-Walter Rix. Galactic masers and the Milky Way circular velocity. *Astrophys. J.*, 704:1704–1709, 2009.
- [23] E. R. Priest. *Solar magneto-hydrodynamics / Eric R. Priest*. D. Reidel Pub. Co. ; Sold and distributed in the USA and Canada by Kluwer Boston, Inc Dordrecht, Holland ; Boston : Hingham, MA, 1982.
- [24] MARIACHIARA Falco, Giovanni Puglisi, SL Guglielmino, P Romano, I Ermolli, and F Zuccarello. Comparison of different populations of granular features in the solar photosphere. *Astronomy & Astrophysics*, 605:A87, 2017.
- [25] J Bahng and M Schwarzschild. Lifetime of solar granules. *The Astrophysical Journal*, 134:312, 1961.
- [26] Landi, E. Fe xiii line intensities in solar plasmas observed by serts. *AA*, 382(3):1106–1117, 2002.
- [27] Jeffrey W Brosius, Joseph M Davila, Roger J Thomas, and Brunella C Monsignori-Fossi. Measuring active and quiet-sun coronal plasma properties with extreme-ultraviolet spectra from serts. *Astrophysical Journal Supplement v. 106, p. 143*, 106:143, 1996.
- [28] Jeffrey W Brosius, Joseph M Davila, and Roger J Thomas. Solar active region and quiet-sun extreme-ultraviolet spectra from serts-95. *The Astrophysical Journal Supplement Series*, 119(2):255, 1998.

- [29] E Landi and M Landini. Temperature and density diagnostics of quiet sun and active regions observed with cds nis. *Astronomy and Astrophysics*, v. 340, p. 265-276 (1998), 340:265–276, 1998.
- [30] Jack B Zirker. *Sunquakes: Probing the interior of the Sun*. JHU Press, 2003.

REVIEWER COMMENTS

Reviewer #1 (Remarks to the Author):

I thank the authors for their extensive response to each of the concerns that I, and the other referees, have raised. At this point my concerns have been addressed at sufficient level, and I am convinced many of the assumptions adopted throughout this work seem to only have a very marginal impact on the final result. I am happy at this point to recommend this manuscript for publication.

Reviewer #2 (Remarks to the Author):

Comments on feedback from An et al 2023

Firstly it is good to see that the authors have made significant changes to the text. In the absence of any mention, the reviewer is pleased with the changes the authors have made in response to the comments. Below are two areas that the authors still need to address.

2.4 Coronal Density Model

It is good that the authors have compared their density model with one derived from the quiet Sun using LOFAR data. It is a shame that this figure has not made it into the paper itself. However, there is a modification that needs to be made for the density model used in the paper. The value of g_{sun} has been taken at the solar photosphere. In reality, the value of g_{sun} varies as a function of distance from the Sun ($1/r^2$). Whilst this effect is not significant for small coronal loops (and is often approximated using the value at the solar surface), the distances of 1 solar radii that are related to the profile adopted in the paper will have non-trivial effects on their density model. Secondly, a value of 1 MK would be more accurate for the quiet Sun. Indeed, the temperature in the solar corona decreases as a function of distance from the Sun – a natural consequence when the value of g_{sun} varies with distance, and is shown in Figure 12 of Vocks et al 2018. Thirdly, the difference between the adopted density model and the density model derived from LOFAR data looks to be larger than a factor of two. This would be better observed on a log y-axis. What is important for the study is the frequencies between 30-80 MHz and the authors should indicate the relevant densities on the plot.

The authors should consider using the density model provided by the LOFAR observations. Alternatively, the authors can compare/use other typical models like the background coronal models based on white light data near solar minimum (Saito, Poland, and Munro, 1977), which also has an analytical form as requested by reviewer 3.

2.9 Bad Channels

The curve looks much better with the extraction of the bad channels that were corrupting the sub-band intensity. The reviewer notes that these bad channels are likely present in the April 25 around 32 MHz and 39 MHz, and the July 03 data around 34 MHz and 50 MHz. Whilst these arrays are not used for the upper limits in Figure 3, it would show a more accurate portrayal of the LOFAR data if the bad channels were removed before the curves were plotted in Figure 2.

Reviewer #3 (Remarks to the Author):

The authors have addressed in details all my questions and concerns, and I am now happy to recommend the paper for publication in Nature communication.

Dear editors and referees,

We deeply appreciate your valuable comments and insightful suggestions. Your positive assessment of our study is truly invaluable. We have thoroughly reviewed your feedback and have incorporated revisions into the manuscript as suggested. In the sections below, we provide responses to the specific comments raised in the second referee report.

Replies to Referee 2

1.1). *Coronal Density Model*

It is good that the authors have compared their density model with one derived from the quiet Sun using LOFAR data. It is a shame that this figure has not made it into the paper itself. However, there is a modification that needs to be made for the density model used in the paper. The value of g_{\odot} has been taken at the solar photosphere. In reality, the value of g_{\odot} varies as a function of distance from the Sun ($1/r^2$). Whilst this effect is not significant for small coronal loops (and is often approximated using the value at the solar surface), the distances of 1 solar radii that are related to the profile adopted in the paper will have non-trivial effects on their density model. Secondly, a value of 1 MK would be more accurate for the quiet Sun. Indeed, the temperature in the solar corona decreases as a function of distance from the Sun – a natural consequence when they the value of g_{\odot} varies with distance, and is shown in Figure 12 of Vocks et al 2018. Thirdly, the difference between the adopted density model and the density model derived from LOFAR data looks to be larger than a factor of two. This would be better observed on a log y-axis. What is important for the study is the frequencies between 30-80 MHz and the authors should indicate the relevant densities on the plot.

The authors should consider using the density model provided by the LOFAR observations. Alternatively, the authors can compare/use other typical models like the background coronal models based on white light data near solar minimum (Saito, Poland, and Munro, 1977), which also has an analytical form as requested by reviewer 3.

Reply. We are grateful for the referee’s advice on the modification of the density model for the quiet Sun. Following the referee’s suggestion, we switch to the density model provided by the LOFAR observations [1] and update our limits on the dark photon dark matter parameter space, as shown in Fig. 1 here. Compared to the results using the solar profile from Ref. [2],

adopting the profile observed by LOFAR strengthens our limits by a factor of two in the low-frequency range (below approximately 50 MHz). This enhancement arises from the fact that the radiation power per solid angle $d\mathcal{P}/d\Omega$ is proportional to r_c^2 , which is roughly 1.5 times larger than the value obtained using the previous density profile. Additionally, $|d\omega_p/dr|^{-1}$ is about twice as large as before. As a result, the kinetic mixing parameter ϵ , which is proportional to $1/(r_c\sqrt{|d\omega_p/dr|^{-1}})$, is about half the value of the previous limits. In addition, the limits obtained for the axion case are also updated correspondingly in the manuscript.

Figure 1: 95% C.L. upper limits on the kinetic mixing parameter ϵ for DPDM adopting the density model provided by the LOFAR observations [1].

Moreover, we clarify our statements to address the referee's comments. Firstly, the solar profile we used before is from the work of V. De La Luz, et al. [2], which derived solar profile for the quiet Sun based on the photospheric model from Ref. [3] and the coronal model from Refs. [4, 5]. However, when applying the hydrostatic density model to fit the profile of V. De La Luz, we adhered to the parameters outlined in Ref.[1], which compared the density profile of hydrostatic density model, the r^{-2} density profile and the profile they observed. In their study, they utilized the gravitational acceleration $g_\odot = 274 \text{ m/s}^2$ at the coronal base [1]. Furthermore, concerning the temperature in corona, we also adopted the value presented in their paper, which is approximately 2 MK from the LOFAR observation fitting [1].

In order to see the differences between different density model, we followed

the referee’s recommendation and illustrated them in a log-scale plot, as depicted in Fig. 2. In addition to the density profile by V. De La Luz, et al. [2] and the profile provide by LOFAR observation [1], we have also included the simple hydrostatic profile and the r^{-2} density profile, shown as dashed lines. The frequency range we analysed are indicated by gray region. It can be seen that in our frequency range, the differences in n_e are approximately within factor of two, while the differences in r_c are within factor of 1.5. In addition, following the suggestion of the referee and the editor, Fig. 2 is shown in the main text of the manuscript.

Figure 2: Various density profiles are depicted using different lines. The solid blue line represents the profile derived from LOFAR observations [1]. In comparison, the solid orange line represents the profile provided by V. De La Luz, et al. [2]. The dashed cyan line represents a simple hydrostatic model and the dashed red line represents an r^{-2} profile [1]. The gray shaded region denotes the frequency region around 30 – 80 MHz which our study focuses on.

1.2). *Bad Channels*

The curve looks much better with the extraction of the bad channels that were corrupting the sub-band intensity. The reviewer notes that these bad channels are likely present in the April 25 around 32 MHz and 39 MHz, and the July 03 data around 34 MHz and 50 MHz. Whilst these arrays are not used for the upper limits in Figure 3, it would show a more accurate portrayal of the LOFAR data if the bad channels were removed before the curves were plotted in Figure 2.

Reply. We extend our gratitude to the referee for providing valuable advice. The potential bad channels, which lead to peaks in the left panel of Fig. 3, correspond to the 25th, 26th, and 27th bins ($f \sim 32.2$ MHz), the 34th bin ($f \sim 33.0$ MHz), and the 101st bin ($f \sim 39.5$ MHz) from the observations on April 25, 2015. Additionally, from the observations on July 3, 2015, the 46th bin ($f \sim 34.2$ MHz) and the 208th bin ($f \sim 50.0$ MHz) exhibit potential issues.

Following the exclusion of these frequency bins associated with potential bad channels, the right panel of Fig. 3 displays the 95% confidence level signal upper limits.

Figure 3: Model-independent 95% C.L. signal limits, with the left panel displaying the limits without removing the frequency bins associated with possible bad channels, and the right panel showing the limits after removing possible bad channels.

Summary

We have switched to the coronal density model from LOFAR observations for our analysis, as suggested by the referee. We have added Figure 1 in the main text to compare different density profiles and modified the text correspondingly. We have refined Figure 3 by excluding potential bad channels. Additionally, by incorporating the new density profile and addressing bad channels, we have updated our limits on the dark photon dark matter coupling in Figure 4 of the main text and also the limits on the axion dark matter coupling in Figure 10 of the Methods section.

In addition to the revised manuscript, we have included a diff-PDF file that clearly highlights all changes for easy tracking.

We hope that our responses and adjustments sufficiently address the com-

ments and inquiries outlined in the report, and we eagerly await acceptance of our paper for publication in Nature Communications.

Sincerely yours,

Haipeng An, Xingyao Chen, Shuailiang Ge, Jia Liu, and Yan Luo

References

- [1] Christian Vocks, Gottfried Mann, Frank Breitling, MM Bisi, B Dabrowski, R Fallows, PT Gallagher, A Krankowski, J Magdalenić, C Marqué, et al. Lofar observations of the quiet solar corona. *Astronomy & Astrophysics*, 614:A54, 2018.
- [2] V. De La Luz, A. Lara, E. Mendoza, and M. Shimojo. 3D Simulations of the Quiet Sun Radio Emission at Millimeter and Submillimeter Wavelengths. *Geofisica Internacional*, 47:197–203, July 2008.
- [3] J. E. Vernazza, E. H. Avrett, and R. Loeser. Structure of the solar chromosphere. III. Models of the EUV brightness components of the quiet sun. *Astrophysical Journal, Suppl. Ser.*, 45:635–725, April 1981.
- [4] A. H. Gabriel. A Magnetic Model of the Solar Transition Region. *Philosophical Transactions of the Royal Society of London Series A*, 281(1304):339–352, May 1976.
- [5] Peter Foukal. *Solar Astrophysics*. A Wiley-Interscience publication. Wiley, 1990.

REVIEWERS' COMMENTS

Reviewer #2 (Remarks to the Author):

Dear Haipeng An, Xingyao Chen, Shuailiang Ge, Jia Liu, and Yan Luo,

Thank you for addressing my concerns with regards to the density model and the bad channels in the LOFAR data. I am satisfied with these corrections and believe your paper is ready for publication.

Reviewer #3 (Remarks to the Author):

I am happy with all the modifications made by the authors and in my opinion the manuscript is now ready for publication in its actual form.